# Bacterial Membrane Vesicles as Smart Drug Delivery and Carrier Systems: A New Nanosystems Tool for Current Anticancer and Antimicrobial Therapy

**DOI:** 10.3390/pharmaceutics15041052

**Published:** 2023-03-24

**Authors:** Pınar Aytar Çelik, Kubra Erdogan-Gover, Dilan Barut, Blaise Manga Enuh, Gülin Amasya, Ceyda Tuba Sengel-Türk, Burak Derkus, Ahmet Çabuk

**Affiliations:** 1Environmental Protection and Control Program, Eskisehir Osmangazi University, Eskisehir 26110, Turkey; 2Department of Biotechnology and Biosafety, Graduate School of Natural and Applied Science, Eskisehir Osmangazi University, Eskisehir 26040, Turkey; 3Department of Pharmaceutical Technology, Faculty of Pharmacy, Ankara University, Ankara 06100, Turkey; 4Department of Chemistry, Faculty of Science, Ankara University, Ankara 06560, Turkey; 5Department of Biology, Faculty of Science, Eskisehir Osmangazi University, Eskisehir 26040, Turkey

**Keywords:** bacterial membrane vesicle, smart drug delivery systems, carrier adjuvant systems

## Abstract

Bacterial membrane vesicles (BMVs) are known to be critical communication tools in several pathophysiological processes between bacteria and host cells. Given this situation, BMVs for transporting and delivering exogenous therapeutic cargoes have been inspiring as promising platforms for developing smart drug delivery systems (SDDSs). In the first section of this review paper, starting with an introduction to pharmaceutical technology and nanotechnology, we delve into the design and classification of SDDSs. We discuss the characteristics of BMVs including their size, shape, charge, effective production and purification techniques, and the different methods used for cargo loading and drug encapsulation. We also shed light on the drug release mechanism, the design of BMVs as smart carriers, and recent remarkable findings on the potential of BMVs for anticancer and antimicrobial therapy. Furthermore, this review covers the safety of BMVs and the challenges that need to be overcome for clinical use. Finally, we discuss the recent advancements and prospects for BMVs as SDDSs and highlight their potential in revolutionizing the fields of nanomedicine and drug delivery. In conclusion, this review paper aims to provide a comprehensive overview of the state-of-the-art field of BMVs as SDDSs, encompassing their design, composition, fabrication, purification, and characterization, as well as the various strategies used for targeted delivery. Considering this information, the aim of this review is to provide researchers in the field with a comprehensive understanding of the current state of BMVs as SDDSs, enabling them to identify critical gaps and formulate new hypotheses to accelerate the progress of the field.

## 1. Introduction

Considering that the discovery of a new drug molecule is both costly and time-consuming, the use of state-of-art drug delivery systems is the most up-to-date approach in the pharmaceutical sciences to increase therapeutic efficacy as well as reduce the side-effect profile of currently used active molecules. Therefore, the concept of a drug delivery system keeps up with technological advances. It has also become inevitable for drug delivery systems to become smarter in today’s world, as everything from the watch to the phone has become smart.

In the classical sense, “drug delivery system” refers to the technologies that carry the active ingredients used for the treatment of any disease into the body. The principal purpose of conventional drug delivery systems is to transport the active molecule in the optimal dose and in a convenient way as well as to facilitate patient compliance and to maintain the stability of the active substance. However, the most notable drawback of conventional systems is that they offer low bioavailability due to the physicochemical properties of the active molecule and they may cause fluctuations in plasma drug level. Drug release from these systems cannot be controlled either [1]. The low water solubility and poor permeability of active molecules as well as rapid metabolization and elimination are associated with their bioavailability, and these features largely limit the pharmacological potential of the active molecules. Therefore, advances in the development of drug delivery systems have always been aimed to increase bioavailability. While we cannot intervene in the physical properties of active molecules, all therapeutic functions can be controlled with the construction of an effective delivery system [2,3].

Since the rise of nanotechnology in the early 2000s and its relationship with drug release technologies, the advantages of “nano-size” over bioavailability have been exploited and many studies have been conducted on nanoscale drug release strategies and their PEGylated derivatives. Today, many commercial examples of “nanocarriers” appear in the world pharmaceutical market. This is simply reasoned by the unique advantages of adaptation of nanotechnology to drug delivery systems, such as the increased release of drugs with low water solubility, facilitated access of the active substance to cells or tissues, efficient passage through epithelial and endothelial barriers, the acquired potential of simultaneous application of multiple substances, and the enabled ability to develop theranostic systems by combining the active substances with imaging agents [4].

The expectation from a novel drug delivery system involves a specific delivery of the loaded active molecule only to the damaged/diseased tissues and cells where a pharmacological activity is needed. Beyond any doubt, the aim of this approach is strictly to maximize treatment response and minimize side effects. Thus, besides basic concepts such as drug delivery systems, controlled release, and nanotechnology, materials science also comes into play and contributes to emerging smart drug systems strategies. The smart drug delivery phenomenon is mostly based on materials; in many cases polymers with unique properties that sense a change upon external exposure. Later, they initiate a response to the stimuli via different mechanisms depending on the chemical bases of the building-blocks, in other words, they exhibit stimuli-responsive features. This phenomenon can be adapted to the macro, micro, or nanoscales [5]. Stimuli-responsive drug delivery systems can be created in various organic architectures such as vesicles, polymeric nanoparticles, micelles, dendrimers, or hydrogels as well as inorganic metal oxide frameworks, quantum dots, and mesoporous silica nanoparticles. Basically, the reason why the carrier system is called “smart” is that it reveals the active component it carries as a result of the system’s response to various stimuli. Depending on the source of the stimulus, as endogenous or exogenous, stimuli-responsive materials can be used to effectively deliver the drug of interest [6,7,8,9].

Bacterial membrane vesicles (BMVs) derived from bacteria have also been recognized as smart drug delivery systems due to their biomimetic properties and versatile functions, as well as the nature of the vesicles, which are important in intercellular communication [10]. They are distinguished from other nano-sized drug delivery systems as next-generation biomimetic vehicles that can be produced on a large scale at lower costs. Another advantage they provide is that isolated BMVs are amenable to surface modification and allow active targeting via specific ligands, or it is possible to obtain tailor-made BMVs by genetic manipulation on vesicle source strains. Moreover, the most fundamental feature of BMVs that enable them to be used as smart drug delivery systems is their ability to evade the host immune system and maintain the stability of the therapeutic agent [11].

Herein, we have focused on the roles and potentials of BMVs in smart delivery systems; additionally, the fabrication principles, characterizations, and applications of BMVs have been covered.

## 2. Design and Classification of Smart Drug Delivery Systems

### 2.1. Design

The main purpose of drug delivery systems is to deliver and target drugs to tissues in a protected manner. The drug given with traditional systems is quickly eliminated from the body. In these cases, regular multiple doses seem to be an alternative, but toxic levels can be observed as a result of overdosing, resulting in poor patient compliance. Smart drug delivery systems (SDDSs) are required to obtain a steady-state concentration in targeted tissues to provide a prolonged effect and eliminate side effects. While designing these systems, there are important parameters such as biomaterial properties, stability, and stimulants. Delivery systems sensitive to stimuli are grouped into two categories: those sensitive to internal and external stimuli [2].

#### 2.1.1. Internal

Among the internal stimulants, there are pH-responsive, redox-responsive, enzyme-responsive, and ionic microenvironment-responsive systems. In SDDSs designed for stimulation by internal stimuli, drug release occurs with the deterioration of the physicochemical structure of the materials suitable for these stimuli [12].

pH-responsive. Among the internal stimulants, the pH factor is frequently used [13]. In pH-responsive biomaterials, deterioration occurs that triggers drug release according to the difference in pH. While in the normal pH range, pH-responsive carriers keep the drug stable and release it with the changing pH in the therapeutic tissues after depots [2]. Ionizing polymers such as polyacrylic acid and polymethacrylic acid can be used to design pH-responsive carriers. These pH-responsive polymers are categorized as polyacids and poly basics [14,15]. 

Redox-responsive. While the intracellular reduced glutathione (GSH) concentration is between 2 and 10 mM, this ratio is 1/1000 outside the cell. Thus, a redox gradient is formed between the extracellular and the intracellular. The disulfide bond has been recognized as the main binding agent for redox-sensitive systems. While the disulfide bonds in the transporters are stable at low GSH levels in the extracellular environment, they are reduced to thiol groups at high GSH levels inside the cell, and the transporters are disrupted resulting in drug release [14]. In addition, considering the high accumulation of reactive oxygen species (ROS) in some therapeutic situations, carrier systems have been developed to respond to ROS [13,16].

Enzyme-responsive. In certain therapeutic situations, some enzymes are specifically produced at high levels. By using these specific enzymes, SDDSs with high substrate specificity and selectivity have been developed [13,17,18].

Ionic microenvironment-responsive. Ionic microenvironment-responsive carriers are designed by adding acidic and basic functional molecules that affect the ionization power. The presence of highly acidic groups in the carriers creates increased electrostatic repulsion between the negatively charged groups, and at high pH, the physiology of the carrier changes. Thus, drug release is triggered [12].

#### 2.1.2. External

In the class of external stimulants, some systems are temperature-responsive, light-responsive, electrical field-responsive, magnetic field-responsive, and ultrasound-responsive systems.

Temperature-responsive. Among carriers sensitive to external stimuli, temperature-responsive ones are preferred as a more common strategy. The applicability of naturally occurring and easily generated temperature differences is the main reason for this [16]. The presence of high temperatures in therapeutic tissues, ranging from 40 to 45 °C, makes the use of temperature-responsive carriers important [19]. Basically, temperature-responsive carriers retain the drug at a normal temperature and release it when exposed to the high temperature of the therapeutic tissue [12]. There are two types of temperature-responsive carrier systems. These are the low critical solution temperature (LCST) and the upper critical solution temperature (UCST). In the LCST type, the carrier physiology is triggered at temperatures below the LCST; in the same way, in the UCST type, the carrier physical chemistry is triggered at temperatures above the UCST and directs the drug release [12,18].

Light-responsive. Ultraviolet light, visible light, and near-infrared light (NIR) are external stimuli used for photosensitive carriers. NIR, with its advantages such as high biocompatibility and in situ polymerization, is an effective stimulant as a photosensitive drug carrier. Different mechanisms have been reported for drug release by systems that respond to NIR: the photo-thermal effect and two-photon activation [12,16,19].

Electrical field-responsive. Heat generation and redox reactions occur as a result of the electrical application in drug delivery systems designed as sensitive to an electric field. In this way, different drug-release pathways are activated by electrical stimulation. Electrical field-responsive polymers such as polypyrrole, polyaniline, and graphene are examples of this system [2,12].

Magnetic field-responsive. The magnetic field can penetrate tissues and is frequently used in imaging. Apart from imaging, it has potential as an effective external stimulant for drug delivery systems. Magnetic field-responsive drug delivery systems have magnetic field-induced hyperthermia and magnetic field-directed drug targeting mechanisms. When a magnetic field is applied, heat is generated. Therefore, magnetic nanoparticles are encapsulated in colloidal carriers that trigger drug release in the case of hyperthermia in magnetic field-responsive systems [2,12].

Ultrasound-responsive. Ultrasound waves are used as external stimulants in drug release because of their tissue penetration and good spatio-temporal control properties. Ultrasound waves create thermal, mechanical, and radiation forces and these effects trigger drug release in carrier systems [12].

Dual/Multi-responsive. In addition to drug delivery systems responsive to a single stimulant, combinations of multiple internal or external stimulators can be used to increase efficiency. Among systems responsive to dual stimuli, pH/temperature-responsive systems have been extensively studied. These systems are used in situations that are difficult to target when only temperature-responsive systems are used [15,16].

### 2.2. Classification

There are three main release mechanisms for drug delivery systems: diffusion controlled, osmotically controlled, and chemically controlled. In diffusion-controlled systems, the drug is retained in water-insoluble polymeric membranes or matrices and released by diffusion. Biocompatible membranes with water permeability are used in osmotically controlled systems. In these systems, osmotic pressure is created by the use of NaCl or formulation with an osmogenic effect. Polymers such as cellulose acetate and ethyl cellulose are widely used in osmotically controlled systems. In chemically controlled systems or, in other words, erosion-controlled systems, materials that can degrade in a biological environment are used. There are two types of mechanisms in these systems: polymer-drug dispersion and polymer-drug conjugation. In polymer-drug dispersion, the drug is dispersed into the biodegradable polymer, and drug release occurs with the degradation of the polymer under therapeutic conditions. In the polymer-drug conjugate, the drug is conjugated to the polymer surface by covalent bonds, and drug release occurs by breaking the polymer-drug bonds under therapeutic conditions [2].

Nanosized drug carriers have a high loading capacity, increase bioavailability by keeping the drug in circulation for longer periods, and are advantageous for targeted delivery as their surface is easy to modify [2]. Many nanosized carriers are available for SDDSs.

Mesoporous silica systems. Mesoporous silica systems range in size from 50 to 300 nm. Endocytosis allows them to enter the cell without causing cytotoxicity. It has inner and outer surfaces, and these surfaces can be modified to make it functional [20].

Hydrogels. Hydrogel structures are frequently used in drug delivery. They consist of water-soluble polymers with cross-linked networks. The release of the drug embedded in the hydrogel occurs by the swelling of the polymer in the aqueous medium. Temperature, pH, and the ionic environment are effective in swelling the hydrogel [18].

Dendrimers. Dendrimers are less common as carrier systems. They have a central core, an inner layer of blocks, and a peripheral region consisting of hyperbranched chains. They can be functionalized with specific ligands to form SDDSs. Its advantages include providing controllable size and molecular weight [20,21].

Liposomes. Liposomes are composed of phospholipids arranged in vesicle form. Their size ranges from nanometer sizes to micrometer sizes. Vesicular systems inspired by liposomes can be categorized as niosomes, transfersomes, etosomes, and phytosomes. Niosomes have a non-ionic surface containing low concentrations of phospholipids. Transfersomes are liposomes that have increased flexibility by adding a single chain activator to the surface. If ethanol is used in the preparation of liposomes, they are called etosomes. Finally, if the phospholipids used in the composition are obtained from plants, the structures formed are called phytosomes. The liposome core is suitable for encapsulating hydrophilic drugs, while the peripheral region between the lipid and phospholipid layers encapsulates hydrophobic drugs [2,21].

Metal nanoparticles. Metal nanoparticles can be used in the visualization and diagnosis of cellular components as well as in drug delivery. Gold and silver nanoparticles are the most popular. They can be functionalized with bioactive molecules such as antibodies and enzymes. These structures can be included in SDDSs by adding hybrid features [20].

Polymeric nanoparticles. Polymeric nanoparticles are usually 100–700 nm in size and consist of biodegradable polymer/copolymer structures. Drugs can be encapsulated in these structures, physically absorbed at the surface, or chemically bound to the surface [20].

Solid-lipid nanoparticles (SLN). Solid lipid nanoparticles have emerged by combining polymeric nanoparticles and liposomes. They can be defined as particles in the nanometer range consisting of biocompatible lipids in solid form at room/body temperature [2].

Carbon nanotubes (CNT). Carbon nanotubes consist of spherical graphene layers that can be sealed with fullerene. CNTs must be functionalized in order to be used in biological systems; otherwise, they may show cytotoxic effects. Bioactive agents can be conjugated or coated with biodegradable agents to functionalize them [2,20].

Polymeric carbon nanoparticles. These are formed by modifying carbon nanotubes with polymers. Polymer wrapping or polymer adsorption can occur based on weak wavelike interactions between carbon nanotubes and polymers, or polymers can be integrated into the carbon nanotube surface by chemical bonds [20].

Exosomes. Exosomes are vesicles derived from eukaryotic cells that range in size from 30 to 100 nm. It is a biological material and has been frequently studied in recent years due to its ability to enter cells. The advantages of exosomes are their non-cytotoxic, natural targeting abilities and high drug-loading capacity. The disadvantages of exosomes are the difficulty in their large-scale production and purification without losing their biological properties in clinical use [2,13,22].

Bacterial membrane vesicles (BMVs). BMVs are vesicles derived from bacterial cells. BMVs have certain advantages over existing carrier vehicles: they are more stable against leakage and deterioration by keeping the drug more stable in circulation. They can increase the effect of the drug with their immunogenic effects due to the presence of antigen and LPS on their surface. They are highly suitable for surface modification and genetic engineering regulations that will increase targeting efficiency [23].

Exosomes from eukaryotic cells and membrane vesicles from bacteria have numerous structural and biologically functional similarities [24]. Although exosomes are being investigated more and more as carrier systems with high biocompatibility and translation ability around the world, their manufacture is challenging and costly, especially in large-scale mammalian cultures, when compared to BMVs. On the other hand, the production of BMVs in bacterial cultures is faster, simpler, and relatively cheaper. In addition, BMVs are nanostructures specifically suitable for design and production, where genetic engineering is more convenient [11]. Although research on exosomes in biological applications is intense, investigations in this field have increased steadily in recent years after the discovery of BMVs [25].

The basic features of BMVs, their production and purification techniques, and their use as SDDSs in the biomedical field will be explained in detail in the chapters of this review paper. The classification of SDDSs as well as their advantages and disadvantages are presented in Table 1 [22,26,27,28,29,30,31,32,33,34,35,36,37,38,39,40,41,42,43,44,45].

## 3. BMVs as Smart Drug Delivery Systems

The history and timeline of milestones of BMVs and their applications in drug delivery can be traced back to the late 1970s and early 1980s when the concept of using BMVs as drug carriers was first proposed. A general timeline of key events and milestones in the history of BMVs and their applications in drug delivery is shown in Figure 1. While BMVs show great promise as drug delivery systems, there are still concerns about their stability and toxicity. However, one novel angle to consider is the potential for BMVs to be modified to respond to specific environmental cues within the body, such as pH or temperature changes, to release their contents. This “smart” approach could increase drug efficacy while reducing potential side effects.

### 3.1. Fabrications of BMVs

The release of extracellular vesicles is an evolutionarily conserved feature. They are produced from the cells of all organisms, including prokaryotes and eukaryotes. The vesicles produced by bacteria are generally referred to as “bacterial membrane vesicles”. BMVs are defined as non-replicative spherical nanostructures with a lipid bilayer about 100–400 nm in diameter [46]. The discovery of BMVs dates back to the 1960s with the development of the electron microscope [47].

BMVs are released from both Gram-negative and Gram-positive bacteria. Basically, three models are presented for BMV biogenesis. The first is through the loss of outer membrane and peptidoglycan layer connectivity by the deposition of outer membrane proteins in the periplasmic space. Evidence for this model is the reduction of outer membrane proteins in vesicle-producing bacteria. In the second model, it is assumed that the turgor pressure created by the accumulation of periplasmic proteins and peptidoglycan fragments in the periplasmic space may cause vesicle formation. Finally, vesicle formation is believed to be facilitated by the conformational change of phospholipids on the membrane, leading to a curvature in the membrane [48].

BMV production has been shown to be impacted by various stress factors, including the growth environment, oxidative stress, and temperature. Studies have revealed an increase in the number of BMVs when bacteria are grown in nutrient-enhanced environments and acidified growth media. Similarly, when oxidative stress was introduced through a non-lethal dose of hydrogen peroxide to *Pseudomonas aeruginosa*, an increase in the number of BMVs was observed. Heat shock, which causes protein denaturation, has also been shown to increase the production of BMVs as these vesicles act to remove these denatured proteins. Furthermore, genetic modifications besides stress factors have also been reported to augment BMV production [48].

### 3.2. Purification and Characterization Techniques of BMVs

BMVs for biotechnological uses must be obtained in high purity and be well characterized. So far, there has not been described a standard method for BMV isolation. Hence, researchers mostly combine multiple methods. Centrifugation and filtration of bacterial cultures is the first step for the isolation of BMVs.

Serial ultracentrifugation is the most widely utilized method for BMV isolation. This approach involves multiple steps to reach BMVs with the highest purity. First, the bacterial culture is centrifuged at approximately 10,000× *g* to remove bacterial cell debris. The resulting supernatant is then filtered through a 0.22–0.45 µm filter, and cellular debris is completely removed. Afterward, the cell-free supernatant is ultracentrifuged at 100,000–200,000× *g* for 1–4 h to obtain BMVs [49]. The disadvantages of this method include long processing times and a loss of integrity in BMVs. In addition, different centrifugal forces and times need to be standardized for this method [50].

Ultrafiltration is another frequently used method in BMV isolation. It is also used to concentrate BMVs following the centrifugation step. This technique provides various advantages such as fast and easy operation, yet it cannot purify the vesicles from other structures with similar size, depending on the pore size of the filter, and therefore may cause contamination. In addition, vesicles may be deformed by the force applied during ultrafiltration [25].

The precipitation method is a reliable technique for removing macromolecules in solution, and it can also be used for purifying BMVs. To achieve optimal results, precipitation is often combined with ultracentrifugation and ultrafiltration methods. Although the precipitation procedure for BMVs requires multiple steps and additional purification due to the saturation concentration, it is still a cost-effective and efficient option thanks to commercially available precipitation reagents. In fact, a commercial kit specifically designed for BMV isolation is now available [51].

The ultracentrifugation method is another of the techniques widely used for the purification and isolation of BMVs. The process involves density gradient centrifugation (DGC), where the samples are loaded into a density gradient medium and subjected to centrifugation. This allows for the separation of BMVs based on their density, making it easier to isolate the vesicles through fraction collection. Iodixanol solution is commonly used in this method as it helps preserve the size and shape of the vesicles. However, the DGC method also has its limitations, such as the potential for compromising vesicle integrity due to changing osmotic conditions and the cost of equipment, which may limit its sustainability [52,53].

Size exclusion chromatography is another alternative method used in the purification of BMVs. This method is based on the collection of particles of varying sizes as they pass through the porous material within the column. As BMVs are too large to interact with the column material, they are isolated quickly and easily. Compared to centrifugation-based methods, the size exclusion method is faster and simpler, and it also preserves the integrity of the vesicles as it does not involve any physical force. However, its disadvantage lies in the limited amount of sample that can be processed due to the column volume, which is why it is often combined with other isolation methods [52,53].

The next important step is their characterization after the pure and efficient isolation of BMVs. The characterization of BMVs can be basically divided into three categories: size, morphology, and composition. For dimensional analysis of BMVs, dynamic light scattering (DLS) and nanoparticle tracking analysis (NTA) are mostly utilized. The diameter, size distribution, and zeta potential of BMVs can be measured simultaneously with DLS analysis. DLS analysis is advantageous in terms of measurement speed and repeatability. With NTA analysis, both the size characterization of BMVs and the number of particles that reflect the yield capacity can be calculated. The advantage of this analysis is that it can measure even small (50 nm) vesicles [54,55].

The structures of BMVs can be investigated using various microscopic techniques. Scanning electron microscopy and transmission electron microscopy are common methods; however, staining and fixation during imaging can alter the morphology of BMVs. Cryo-electron microscopy is an alternative technique that enables the preservation of the BMV structure and the determination of its total number. BMVs can also be visualized using atomic force microscopy (AFM), which does not require additional imaging but has a limited imaging range [55,56].

Identifying the biomolecules involved in the BMV’s structure is the final step in their characterization. The presence of proteins in BMVs can be identified through qualitative and quantitative analyses, including bicinchoninic acid assay (BCA), sodium dodecyl sulfate-polyacrylamide gel electrophoresis (SDS-PAGE), Western blotting, and mass spectrometry. The BCA is used to determine the total amount of protein, while SDS-PAGE provides a qualitative analysis. Western blotting is employed to detect the presence of specific desired proteins in BMVs. Mass spectrometry, with its high efficiency in proteomic analysis, is utilized to examine the biogenesis and physiological functions of BMVs [57,58]. Moreover, using a matrix-assisted laser desorption ionization (MALDI) based mass spectrometry approach, various fatty acids contained in BMVs can also be characterized under lipidomics studies [59].

### 3.3. Properties of BMVs 

BMVs are classified based on the gram characteristics of the bacteria from which they are released. Outer membrane vesicles (OMVs) refer to vesicles produced by Gram-negative bacteria, while membrane vesicles (MVs) refer to those produced by Gram-positive bacteria. OMVs range from 30 to 250 nm in diameter and share components with the bacterial outer membrane. OMVs have been found to contain proteins, nucleic acids, and lipids, with various functions attributed to the identified proteins [60,61]. On the other hand, MVs range in size from 10 to 500 nm and contain proteins, nucleic acids, and lipids, similar to OMVs [59,62].

BMVs have a wide range of functions, which depend on the biomolecules they carry, including adaptation to the environment and pathogenesis. They play a role in transport systems, biofilm formation, antibiotic resistance, phage removal, microbiota protection, gene transfer, pathogenesis, and immune modulation [59,62] and also are known to transport various biomolecules, including proteins and virulence factors, between cells. One of the most important advantages over other transport systems is their ability to transport more than one type of biomolecule simultaneously and to transmit these molecules over long distances by protecting them from lytic enzymes [63]. This property of BMVs makes them particularly effective in the transmission of toxins and virulence factors, which are key factors in bacterial pathogenesis [64]. The ability of BMVs to carry enzymes such as β-lactamase and drug-binding proteins, and to bind antibiotics outside the cell also contributes to antibiotic resistance in bacteria [65]. Moreover, BMVs can protect bacteria from phage infection by irreversibly binding phage receptors to their surfaces, thus preventing the phages from reaching the bacterial cells [66]. Overall, BMVs play a critical role in bacterial pathogenesis and antibiotic resistance, and, in a way, these mechanisms can be harnessed for the development of novel antimicrobial therapies. BMVs are known to contain both chromosomal and plasmid DNA, which they can protect and transfer against thermal and enzymatic denaturation [67]. These vesicles also play a crucial role in biofilm formation by facilitating interactions between bacterial cells, delivering nutrients, and dispersing on biofilm surfaces [68]. Interestingly, studies have also shown that BMVs belonging to biofilm-forming bacteria can trigger biofilm formation in non-biofilm-forming bacteria [69,70]. Moreover, BMVs can initiate signaling within the host cell through the molecules they carry and contain on their surfaces, creating an immune response. The specific immune response elicited depends on the bacteria from which the BMVs are obtained and the molecules they carry [71]. Finally, it is worth noting that the microbiota, the bacteria that make up the body’s natural microbial community, also produce BMVs. These vesicles have been implicated in the protection of microbiota homeostasis against inflammatory bowel diseases [72].

### 3.4. Cargo Loading or/and Drug Encapsulation Techniques into BMVs

There are two main encapsulation approaches for loading cargo into BMVs. These are energy-dependent (active loading) and energy-independent (passive loading) techniques developed based on the energy requirement principle (Figure 2). The application fundamentals of these different drug-loading techniques are discussed in detail below with specific examples.

#### 3.4.1. Active Cargo Loading 

The active cargo loading principle is an energy-requiring approach and is implemented by electroporation, co-extrusion, and sonication methods. Among these techniques, electroporation is based on the principle of creating pores by breaking the integrity of the membrane in order to create a temporary permeability state in the bacterial cell membrane. For this purpose, short-term electrical voltage pulses are applied to the bacteria cells [73,74]. The pores formed in the cell membrane allow the loading of various chemically synthetic drug compounds. By changing the duration and intensity of electrical voltage pulses applied to the cell membrane, low molecular weight synthetic agents, as well as high molecular weight compounds such as nucleotides, can be introduced into the vehicles. After a certain time, the cell membrane returns to its original structure without any damage. This process has also been utilized to entrap active molecules into other vehicles such as exosomes that have a lipid bilayer membrane structure similar to BMVs [75,76].

The electroporation technique was utilized by Gujrati and his co-workers [77] for the purpose of loading cargoes into Gram-negative *Escherichia coli* OMVs. They accomplished inserting fluorescent dye-labeled siRNAs into BMVs by using 50 µF and 700 V electroporation conditions, and without causing permanent damage to the cell membrane integrity. Such probes demonstrate the feasibility of BMVs as theranostic agents in cancer diagnosis and treatment. It is also possible to load nanoparticles such as quantum dots, silver or gold nanoparticles, and iron oxides into BMVs by using electroporation for different therapeutic indications. The prominent need is to optimize the applied electroporation voltage to reach high encapsulation efficiency, optimum vesicle diameter, and stability of the system during the loading of cargo molecules into BMVs [78].

The ultrasonication method, which is another technique utilized for loading cargo molecules into BMVs, is based on the reversible disruption of cell membrane integrity through ultrasonic frequencies, and, thereby, the realization of drug loading [10]. This technique also enables coating the surfaces of conventional polymeric nanoparticles with BMVs to gain biomimetic character [79]. In 2019, Gao and his colleagues coated vancomycin and rifampicin-encapsulated poly(lactic-co-glycolic acid) (PLGA) nanoparticles with BMVs isolated from *Streptococcus aureus* or *E. coli* bacteria by using the ultrasonication technique to impart active targeting ability to the vehicle. Herewith, the researchers improved the in vivo antibacterial activities of the polymeric nanocarriers [80].

The co-extrusion technique is a relatively current method among the active cargo loading approaches. This technique uses the cell membrane supplied by the host and follows a repeated mechanical extrusion process through polycarbonate filter membranes with different pore sizes [10]. Similar to the ultrasonication technique, this drug encapsulation approach is also preferred for the surface modification of various nanosized drug carriers to obtain biomimetic structures. For example, Chen and co-workers modified the surface of tegafur-loaded polymeric micelles with OMVs isolated from *Salmonella typhimurium* to bring efficacy for cancer immunotherapy and to alleviate metastasis. To this goal, they applied two different extrusion steps; in the first step, the immunogenicity of the BMVs was reduced by modification with polyethylene glycol (PEG) and Arg-Gly-Asp peptide (RGD) through the polycarbonate membrane extrusion technique. This step was followed by a second extrusion step which was implemented to coat the micelle surface with BMVs to depress immunogenic activity [81].

#### 3.4.2. Passive Cargo Loading 

Another approach harnessed to load cargo molecules into BMVs is the incubation of drugs with BMVs under appropriate conditions. Kuerban et al. (2020) developed an effective strategy for the treatment of non-small-cell lung cancer with this technique. They aimed to encapsulate doxorubicin hydrochloride (DOX), which is a broad-spectrum antineoplastic agent, into OMVs derived from Gram-negative *Klebsiella pneumoniae*. The incubation procedure for loading the active agent in phosphate-buffered saline (PBS) solution into OMVs was performed at 37 °C for a period of 4 h. To remove the unloaded DOX from the incubation medium, an ultrafiltration step was performed through 100 kDa membranes. The DOX:OMV mass ratio of 1:45 was determined as the optimum ratio to obtain the highest encapsulation efficiency. Particle size analysis demonstrated that the mean size of DOX-loaded OMVs was approximately 93 nm. In vitro dissolution studies depicted that DOX was released from OMVs by 30% over 48 h, exhibiting an extended-release profile [82]. In a study conducted by our research group, Capacitabine-loaded MVs were produced to investigate the anti-apoptotic effects of vehicles derived from Gram-positive *Enterococcus faecalis* on HT-29 colon cancer cell lines. Among the evaluated drug:MV mass ratios of 1:20, 1:40, and 1:60, a maximum loading efficiency of 54% was obtained with a drug:MV ratio of 1:60. Particle size of the optimum formulation was determined as 193.3 nm with a −22.4 mV surface charge. Cell culture studies proved that the drug-loaded MVs increase the apoptosis of cancer cells [83].

Another incubation-based strategy utilized for the loading of cargo molecules into BMVs is to incorporate drugs into BMVs during their biogenesis by the parent bacteria. This technique is mostly preferred for loading mRNA-type genetic materials or aminoglycoside-structured antibiotics that cannot be successfully encapsulated into BMVs after their isolation due to their high molecular weight [10]. Gentamicin was successfully loaded into the OMVs during the growth of *P. aeruginosa* PAO1 by using this technique [84]. However, the major limitation of this approach is the co-transfer of genetic material during the encapsulation process of the antibiotics. Essential purification steps and quality controls have to be carried out after the cargo loading procedure to make sure that genetic material is not entrapped simultaneously with cargo molecules.

### 3.5. Drug Release Mechanisms from BMVs

The main release mechanisms by which the release of the active molecule can occur in drug delivery vehicles are diffusion, dissolution, or erosion. It is important to know the mechanism of drug release from BMVs in order to maximize the expected therapeutic response from the carriers and to expand their clinical use. “*Drug release*” describes the process by which active molecule solutes can pass the membranes of the BMVs to the dissolution medium or biological fluid. Since the outer membrane structures of BMVs contain lipopolysaccharides, proteins, and lipids, the essential mechanism underlying the release of drugs from BMVs is the diffusion mechanism [85]. Huang et al. (2020) produced levofloxacin-loaded BMVs and examined the therapeutic efficacy of these drug-delivery vehicles in a mouse model of intestinal bacterial infection. The diffusion rate of the levofloxacin from orally administered drug-loaded BMVs was observed to be slower than the free molecule. Due to this situation, the drug concentration in plasma was obtained at a lower level but in a more prolonged release type compared to free levofloxacin [86]. However, more research needs to be conducted to elucidate the release kinetics and mechanisms of active molecules from BMVs.

### 3.6. Strategies Used in BMVs for Targeting

As can be seen from the studies in the literature, we will mention two main targeting strategies using BMVs. These are static and dynamic targeting strategies (Figure 3), which will be explained in detail below.

#### 3.6.1. Static Targeting

In this section, we describe passive and active transport, which are among the targeting strategies of tumor tissues. These are referred “*static targeting*” since the operator has no control over the nanosystems injected using these targeting strategies.

##### Passive Targeting

The process of angiogenesis which is necessary for metastasis to occur in tumor development is critical. The development of this process in tumor cells is quite rapid. The endothelial cells of the tumor tissues are not wrapped by pericytes, which ensures the preservation of normal blood vessels, and there is no smooth layer of muscle on the edges of the vessels. Because of this, tumor tissues contain extremely permeable vascular structures also called “*leaky vessels*” [89]. Many solid tumors have excessively permeable vasculature and less lymphatic drainage compared to normal tissues [90]. Enhanced permeability and retention effect (EPR) which is commonly seen in tumorous areas is characterized by increased capillary permeability in affected tissues with much less return of fluids to the lymphatic circulation and is the main driving power of “*passive targeting*” [91]. While free therapeutic agents can spread non-specifically, a nanocarrier system can accumulate in tumor tissues thanks to leaking vessels with EPR [92]. Due to the lack of a functional lymphatic drainage system, nanocarriers cannot be efficiently cleared from the bloodstream, so their circulation in the bloodstream is prolonged [89]. 

Researchers have used BMVs to deploy therapeutic agents through passive targeting. Kuerban et al. showed that DOX-loaded OMVs, due to their nano size, can passively accumulate in lung cancer cells via the EPR effect and then be rapidly internalized by recipient cells by endocytosis. Thanks to the rapid entry of doxorubicin into cells, drug loaded OMVs caused intense cytotoxic effects [82]. In another study, it was emphasized that optoacoustic signals increased significantly in tumor cells due to melanin-loaded OMVs, and this increase was thought to be due to the accumulation of melanin-loaded OMVs in tumor cells via the EPR effect [93].

Although EPR through passive targeting provides an advantage for accumulation in tumor tissues, it is significantly affected due to the heterogeneous nature of tumor tissues, and vascular permeability may not be the same throughout a single tumor [92,94]. To overcome this limitation, BMVs need to be designed to actively bind to specific cells.

##### Active Targeting

Since the application site of therapeutics is usually distant from the sites where the therapeutic agent will be affected, the development of effective targeting strategies is critical to therapeutic effectiveness [95]. In this regard, the research focused on the surface modification of BMVs has opened a new era in SDDSs using biological ligands [77,96,97]. Modified BMVs increase affinity and facilitate the internalization of therapeutic cargo by target cells via receptor-mediated endocytosis and/or disruption of cellular function thanks to an “*active targeting*” strategy based on the biological interaction between ligands and the target cell [98]. 

Even if the nanosystems are intended for active targeting, a passive accumulation occurs first followed by target-specific binding as a complementary strategy [99]. Active targeting increases the effectiveness of cargoes loaded into nanosystems and makes therapeutic cargoes more specific to the target area [100]. Thus, undesirable, non-specific interactions and localization of therapeutic cargo in peripheral tissues are reduced [101]. Herein, the key challenge is to identify the most appropriate targeting ligand for the selective and successful delivery of nanosystems to the target tissue or cell and to minimize or prevent the toxicity that may result [101]. At this point, in the design of the SDDS, the target ligands are selected to bind to a receptor that is overexpressed by the target cell but not by normal cells. Moreover, these targeted receptors must be expressed homogeneously in all targeted cells [102]. 

Biological ligands such as proteins, carbohydrates, nucleic acids, and peptides have been used to facilitate the active targeting of BMVs [87,96,97,103] (Table 2). Human epidermal receptor 2 (HER2) has been found to be expressed in 14–91% of breast cancer patients [102]. Gujrati et al. modified the surface of OMVs obtained from *E. coli* with the HER2 affibody by fusion with the ClyA protein. It showed high specificity of modified OMVs in HER2-overexpressing ovarian (SKOV3) and breast cancer (BT474 and HC1954) cells compared with HER2-negative breast cancer (MDA-MB-231) cells. They also reported that the siRNA molecule transported by this active targeting is effectively transported to the target site and causes more cytotoxicity in cancer cells [77]. Guo and colleagues modified the surface of the OMVs with mannose to target M2 macrophages. They reported increased secretion of recruited tumor-associated macrophages (TAMs) in 4T1 breast cancer cells thanks to increased macrophage affinity compared to native OMVs as the modification rate of mannose-OMVs increases [87]. Aptamers are short chains of nucleic acids consisting of several nucleotides. Due to their small and sensitive structure, biodegradability, and immunogenic effect, they are very good candidates as active targeting ligands for BMVs. AS-1411 is a Guanine-rich DNA aptamer that specifically recognizes the nucleolin that is upregulated in many cancer cells [95]. In a recent study, AS-1411 aptamer was modified by adding to the surface of *E. coli* OMVs and by specifically targeting cancer cells. They also reported that modified OMVs accumulated in tumor tissues and inhibited tumor growth [103].

#### 3.6.2. Dynamic Targeting

To increase therapeutic efficacy and reduce or avoid adverse effects, drugs must be administered to target sites in a controlled manner. Stimuli-based drug delivery systems have demonstrated significant potential for the effective targeting of active drug moieties in this approach [109].

In this section, we will provide insights into dynamic simulations in BMVs used as stimuli-triggered drug delivery systems for cancer therapy through active and passive targeting.

##### Stimuli-Responsive Targeting

To improve drug delivery specificity, efficacy, and biological activities, stimuli-responsive nanocarriers were rationally designed and developed by considering different pathological profiles in normal tissues, intracellular compartments, and tumor microenvironments. Furthermore, it has been reported that stimuli-responsive nanocarriers can overcome multidrug resistance in cancer treatment [110]. Starting from this, research with BMVs has focused on different internal and external stimuli to trigger drug and/or cargo release.

pH. Among the internal stimuli, pH is one of the most used to reduce drug toxicity and provide targeted drug release, especially in oncology and inflammation treatments [19,111]. Tumor cells have an acidic environment due to hypoxia and lactic acid accumulation than normal cells [112]. Based on this feature of tumor cells, pH has been used as an effective stimulus in the controlled release of BMVs.

A new drug delivery system that uses OMVs derived from Gram-negative bacteria to co-deliver paclitaxel (PTX) and regulate the tumor metabolism microenvironment was described. The OMVs were pH-sensitive and released PTX when the tumor pH was at 6.8. The system was also designed to deliver a regulated development and DNA damage response 1 (Redd1)-siRNA. The delivery system was shown to successfully regulate tumor metabolism and suppress tumor growth. The OMVs have the potential for use in establishing a co-delivery platform for chemical drugs and genetic medicines. The study emphasizes the pH-sensitivity of the OMV and its ability to target different cells in the tumor microenvironment successively [87].

Light. Light-responsive drug delivery systems that use photosensitive carriers exhibit an on/off drug release mechanism when stimulated with light. Various light wavelengths (ultraviolet, near-infrared, visible) have been reported and discussed for controlled drug delivery. Because of their low penetration, visible and UV light were deemed unsuitable for clinical purposes in vivo, whereas the NIR spectrum is regarded as an ideal light source for monitoring drug release due to its safety and enhanced tissue penetration [109].

A recent study has focused on loading biomimetic copper sulfide nanoparticles (CuS-OMVs) into OMVs derived from *E. coli* Nissle 1917 for systemic photothermal-immunotherapeutic synergy. CuS-OMVs have high photothermal conversion efficacy, good photostability, and significant tumor targeting capacity, resulting in visible hyperthermia and subsequent distinct suppression of tumor cells when exposed to second near-infrared (NIR-II) light. CuS-OMVs-induced cytotoxicity induces strong immunogenic cell death (ICD) in tumor cells while also promoting dendritic cell (DC) maturation and subsequent CD8^+^ T cell activation [113].

##### Dual/Multi-Responsive Targeting

Dual/multi-responsive targeting of BMVs involves the use of targeting moieties that respond to multiple signals, such as pH, temperature, or enzymes, to achieve selective targeting of BMVs. This strategy has the advantage of allowing for multiple triggers to control the release of drugs from BMVs, thereby increasing the level of control and specificity in drug delivery.

One example of a dual/multi-responsive targeting strategy is the combined use of molecules and light-sensitive molecules that will promote the production of ROS as a result of redox reactions as targeting moieties for BMVs. These molecules can be incorporated into the BMVs, and their response to changes in the redox and light of the environment can be used to trigger the release of drugs from BMVs. This allows for a high degree of control over the timing and location of drug release [114]. A new drug delivery system has been designed that uses αPD-L1-modified OMVs derived from Gram-negative bacteria to deliver the enzyme catalase which breaks down H_2_O_2_ into O_2_, and the photo-sensitive Ce6 molecule together. This nanosystem relieved hypoxia for a long time in vivo by attenuating the hypoxic feature of solid tumors, which inhibited their photodynamic activity. On the other hand, it has been reported that bacterial-based SDDSs developed due to stimuli induces greater DC and CD8^+^ T cell migration into tumor tissues and improves immune memory in treated mice [114].

A study focused on the use of OMVs for TME reprogramming in solid tumors. To overcome the issues of antibody-dependent clearance and high toxicity caused by OMVs upon intravenous injection, the OMVs were covered with pH-sensitive calcium phosphate (CaP) shells. These CaP shells were able to neutralize the acidic TME and reprogram macrophages from the M2-to-M1 phenotype, resulting in an improved antitumor effect. Additionally, the CaP shells could be functionalized with folic acid or photosensitizer agents, allowing for the use of OMVs in combination therapies to achieve a synergistic therapeutic effect. The pH-sensitivity of the CaP shells is emphasized as a crucial feature in the effects delivered by the OMVs [88].

Another example of dual/multi-responsive targeting is the use of enzymes as triggers for drug release. For instance, enzymes that are expressed in diseased tissues can be used to trigger the release of drugs to those tissues. This allows for the targeted delivery of drugs to specific regions of the body while avoiding systemic toxicity. It is a promising strategy for achieving selective and controlled drug delivery, and further research is needed to fully understand the potential of this approach. 

In terms of recent discoveries in the field of dual/multi-responsive targeting of BMVs, one notable example is the use of dual-responsive polymers for the targeted delivery of drugs to cancer cells. Researchers have developed a dual-responsive polymer that can respond to both pH and temperature, and that can be used to achieve targeted delivery of drugs to cancer cells. The polymer is capable of triggering the release of drugs from nanostructures in response to changes in the pH or temperature of the microenvironment of cancer cells, which allows for precise and controlled delivery of drugs to these cells [115,116]. These approaches have been more extensively investigated with liposomes but studies examining their applications in BMVs are lacking.

##### Inverse Targeting

Inverse targeting of BMVs involves the use of targeting moieties that actively avoid specific regions or conditions, rather than being attracted to them. This strategy is useful for avoiding unwanted toxicity or interactions with non-target cells and could be used to improve the specificity and safety of BMV-based drug delivery.

For example, inverse targeting can be achieved using magnetic nanoparticles that are coated with targeting moieties that bind to vesicles. The magnetic nanoparticles are then functionalized with drugs, and their magnetic properties can be used to direct them away from specific regions or conditions. This allows for targeted delivery of drugs to specific regions of the body while avoiding systemic toxicity. Another example of inverse targeting is the use of light-sensitive polymers as targeting moieties. These polymers can be triggered to release drugs upon exposure to specific wavelengths of light. This allows for the precise and controlled release of drugs and can be used to improve the safety and specificity of drug delivery [117]. However, it is important to note that the field of inverse targeting in bacterial membrane vesicles is still in its infancy, and much more research is needed to fully understand its potential benefits and drawbacks.

## 4. The Role of BMV-Based Smart Drug Delivery Systems in Diagnosis

BMVs are candidates with the potential to be used for bioimaging in the early diagnosis of cancer and evaluation of treatment efficacy due to their ability to encapsulate imaging reporters and target ligands. In this regard, Gujrati and colleagues engineered *E. coli* cells to overexpress the tyrosine enzyme, resulting in melanin that was contrast-enhancing by optimizing it as a natural light absorber accumulating in the bacteria’s cytosol and packaged in OMVs released from *E. coli.* Melanin-containing OMVs (OMVMel) capable of generating optoacoustic signals under infrared light induced apoptosis in 4T1 breast cancer cells. In the murine orthotopic breast cancer model, OMVMel exhibited strong necrotic activity by accumulating in tumor tissues with the effect of EPR. These locations have been non-invasively monitored utilizing OMVMel as multispectral optoacoustic tomography (MSOT) probes. Moreover, OMVMel have been demonstrated to be biocompatible without causing any organ damage. As a consequence, when exposed to pulsed near-infrared radiation, OMVMel displayed a theranostic potential by boosting localized heat generation, enabling the suppression of tumor development as well as tumor visualization [93].

Fusion proteins can be attached to membrane-associated proteins on BMVs through genetic modification and serve many different purposes such as protein transport, fluorescent molecular labeling, tumor therapy, and cell bioimaging [118]. The OMV-based multifunctional biosensor platform for both antigen targeted, and signal generation developed by Chen et al. (2017) is notable for its simultaneous functionalization of both the interior and exterior of the OMV. To achieve this, the researchers used SlyB lipoprotein as an anchor to direct nanoluciferase (Nluc) interior *E. coli* OMVs. The choice of Nluc as a fusion partner was based on its small size of 19 kDa and highly sensitive luminescence signaling. In a novel strategy, the OMVs were operationalized with a target-specific antibody using the ice nucleation protein (INP)-Scaf3 surface scaffolding and the small antibody-binding Z domain on the OMV surface. To test the biosensor’s ability to target cancer cells, the researchers chose the cancer-specific surface marker Mucin-1 (MUC1). They further modified the OMVs by mounting dockerin-labeled green fluorescent protein (GFP) for immunofluorescence imaging on the INP-Scaf3-Z scaffold. When GFP-OMVs were added to HeLa cells stained with anti-MUC1 antibody, bright green fluorescence was detected in cells, demonstrating the OMVs’ ability to detect cancer cells. Overall, this study shows that OMVs can be used for immunofluorescence imaging with target-specific antibodies [119]. Another study evaluated the bioluminescence kinetics of OMVs generated from modified *E. coli* using the same genetic alteration method. A theoretical model was developed to simulate the enzyme-substrate reaction kinetics by mixing these OMVs with the substrate furimazine, which could potentially be used for bioluminescence-based optical imaging. OMVs were injected subcutaneously in a mouse model to investigate the local absorption of substrate and OMVs and bioluminescence kinetics. The bioluminescence signal produced by these OMVs was tracked using non-invasive optical imaging. Strong signals from experimental animals indicated that these vesicles have great potential as a class of functional nanomaterials for imaging-related biomedical applications [120].

Presently, there has been limited focus on biosensing and bioimaging utilizing modified BMVs. Because of the increased interest in this topic, numerous OMVs or MVs will be created in the future to be utilized as customized imaging tools with the needed properties through genetic engineering.

## 5. The Potential Role of BMV-Based Smart Drug Delivery Systems in Therapy 

### 5.1. Cancer Therapy

The potential use of BMVs as immunomodulatory candidates in cancer treatment is of great interest because of their multi-antigenic nature [121]. While many studies have reported the anti-tumor effects of OMVs derived from Gram-negative bacteria [113,122,123], there are only a few cancer studies related to MVs obtained from Gram-positive bacteria [83,124,125]. Therefore, this section will mainly focus on the latest research on the use of OMVs in cancer immunotherapy.

According to our literature knowledge, OMVs are utilized in cancer therapy with four different approaches; natural OMVs, cargo and drug-loading OMVs, modified OMVs, and OMVs created with hybrid membrane technology or designed with a coating (Figure 4). Herein, we will examine these nanoscale biomaterial-based carrier systems applied with different strategies as therapeutic and SDDSs and analyze their advantages by summarizing their designs.

#### 5.1.1. Native OMVs

In oncology applications, BMVs offer several advantages. Firstly, their nano sizes ranging from 20 to 200 nm allow for enhanced localization to solid tumors with passive targeting and lymphatic drainage [126]. Secondly, their EPR properties enable them to accumulate in tumor tissue, triggering local immunity [127]. Thirdly, BMVs are enriched in pathogen-associated molecular patterns (PAMPs), such as lipopolysaccharide (LPS), found on their surface and originating from the parent bacteria. This natural adjuvant effect activates various toll-like receptor (TLR) signaling pathways, triggering an inflammatory response associated with the activation of the complement system [48]. Fourthly, BMVs can be genetically engineered through modification of the parent bacteria [93] with molecular techniques and bioconjugation methods for cell targeting and recognition [81,128]. Finally, BMVs cannot reproduce themselves, making them a safer vehicle compared to bacteria [129,130].

According to Kim et al. (2017), membrane vesicles released by different types of bacteria can potentially be utilized for immunotherapy in various types of cancer. *E. coli*-derived OMVs were found to accumulate in tumor tissues via EPR, affecting antigen-presenting cells (APCs) and adaptive immunity, which facilitated the infiltration and accumulation of natural killer (NK) and T cells in tumor tissues. The study further revealed that the antitumor mechanism of OMVs is reliant on interferon-gamma (IFN-γ) production in the tumor microenvironment and CXCL10 expression to attract effector T cells. Additionally, the study was the first to exhibit the substantial anti-tumor effects of MVs produced by Gram-positive bacteria, *Lactobacillus acidophilus* and *S. aureus*, which demonstrates the feasibility of using Gram-positive bacteria for clinical applications [125].

Another study by Aly et al. (2021) found that OMVs obtained from *Salmonella typhimurium* exhibited higher cytotoxicity in cancer cells than paclitaxel (PTX). The researchers utilized OMVs as an adjuvant to enhance the effect of a chemotherapy agent and evidenced that combining OMVs and paclitaxel treatment resulted in a significant reduction in tumor volume and the greatest tumor inhibitory ability at a rate of 94.7% [122].

#### 5.1.2. Cargo or Drug Loading OMVs

OMVs appear to be magnificent platforms in bio-applications due to their unique biological properties; even so, more efforts are needed to take them one step further and improve their functionality for practical use. Long-distance transportation and communication are some core functions of OMVs in biological systems [131,132]. With the aim of improving their therapeutic functions, research has currently focused on their potential as drug-delivery systems [87,133]. OMVs efficiently protect their cargoes against DNase, RNase, protease, and extracellular degradation caused by extreme conditions, such as extreme pH [134]. Additionally, they can carry both hydrophobic and hydrophilic components owing to their lipid bilayer structure [135]. They not only improve drug delivery inside tumor cells but also stimulate immune responses, which boost the effectiveness of anticancer drugs [82,122]. Moreover, OMVs are used in combination with photothermal therapy or photosensitive agents to enhance their immunotherapeutic role to completely eradicate the tumor, prevent recurrence and metastasis [113].

Recent studies have shown promising results in enhancing the antitumor effects against cancer cells by loading the antineoplastic agent DOX into OMVs [82,133]. Li et al. (2023) developed a new smart drug delivery therapeutic platform that co-loaded photosensitizer chlorin e6 (Ce6) and DOX into OMVs for combined photodynamic/chemo/immunotherapy. Then, they loaded the OMVs into macrophages to improve their safety and antitumor effects. The researchers reported that the developed Ce6/DOX-OMVs@M nanoplatform destroyed orthotopic triple-negative breast cancer (TNBC) and prevented lung metastasis in mice with no side effects. The study also demonstrated that laser light could induce ICD by effectively generating ROS, shifting polarization from tumor-associated macrophage M2 to M1, and activating pyroptosis-related pathways to activate antitumor immune responses [133].

#### 5.1.3. Modification of OMVs

Although OMVs play an important role in transporting biomolecules to distant sites, targeting and processing immunomodulatory agents into the tumor microenvironment remains a major challenge. One of the major advantages of using OMVs in cancer therapy is their flexibility to be modified by genetic engineering [127,131]. Modifying them is an excellent strategy to improve the expression of specific peptides and the presentation of multiple antigens in targeting cancer cells and forming the tumor-associated immune response.

The functionalization of OMVs with different biomolecules through genetic engineering has several advantages over other methods: (1) There is no need for decoration material because it is created by introducing heterologous DNA into bacteria. (2) Large-scale production of functionalized OMVs can be achieved at a low cost with simple bacterial cultures. As a result, no further purification steps are required after collection. (3) Finally, this method allows the placement of desired biomolecules inside or outside the OMVs according to different application strategies.

Researchers have explored using various proteins and ligands to enhance the ability of OMVs to target cancer cells. For example, HER2 and peptides (e.g., RGD and RGP) have been investigated as potential targeting moieties [77,96,104]. In one study, tumor necrosis factor-related apoptosis-inducing ligand (TRAIL) protein overexpressing OMVs were produced using recombinant *E. coli* cells, and these vesicles were bonded with RGD peptide which shows high affinity for the most integrin avβ3 that is specifically overexpressed in melanoma cells across the other types of cancers. These nanoplatforms showed strong binding affinity to the integrin receptor, increasing tumor-targeting ability and accumulation in melanoma cells [96].

Recently, OMVs derived from genetically modified basic fibroblast growth factor (BFGF) overexpressing *E. coli* have been shown to be enriched in the BFGF protein [128]. Thus, the immunoadjuvant potential of BFGF-loaded OMVs (BFGF-OMVs) was investigated by their administration to tumor-bearing mice. It was reported that the production of anti-BFGF autoantibodies was induced, tumor metastasis and angiogenesis were inhibited, and a tumor antigen-specific cellular immune response was promoted. Pan et al. (2022) produced OMVs (LOMV@PD-1) that were loaded with plasmids carrying the programmed death-1 (PD-1) and modified with LyP1 polypeptide to induce PD-1 expression in tumor cells. Upon intravenous injection, the LOMV@PD-1 nanostructures were effectively internalized by tumor cells through LyP1 peptide-mediated targeting, and PD-1 expression was observed in tumor cells by transmitting the PD-1 plasmid to the nucleus. The overexpression of PD-1 on the surface of tumor cells inhibited the PD-1/PD-L1 pathway and prevented tumor cells from escaping immunity. Furthermore, OMVs induced IFN-γ secretion, enhancing antitumor immune responses in tumor tissues. The LOMV@PD-1 nanocarrier system exhibited the strongest anti-tumor effect in melanoma cells, with a tumor inhibition rate of 94.21% [105].

Studies on the use of OMVs as carrier adjuvant systems in cancer therapy are summarized in detail in Table 3.

#### 5.1.4. OMVs Designed with Coated and Hybrid Membrane Technology

Nanoparticles (NPs) have gained significant attention as a promising drug delivery system due to their simple design, low toxicity, and in vivo potency enhancement. However, as exogenous substances, they can be efficiently identified and eliminated by the mononuclear phagocyte system (MPS) in the bloodstream, while their nonspecific distribution can lead to increased toxic effects [136]. To overcome these limitations, modified NPs in engineered systems have been proposed as a promising strategy.

In recent years, nanoparticle cell membrane coating technology has emerged as a sophisticated and powerful approach to functionalizing NPs [137]. The approach was initially designed using red blood cell (RBC) membranes to coat polymeric nanoparticles. Subsequently, a broad range of existing cell membrane types such as RBCs, platelets, leukocytes, cancer cells, stem cells, dendritic cells, natural killer cells, and cell membrane-derived structures including exosomes and extracellular vesicles have been comprehensively investigated as nanoparticle coating structures [138].

Researchers are increasingly interested in using OMVs for making NPs more functional in biomedical applications. The coating of NPs with OMVs, referred to as NP-OMVs, preserves the physicochemical properties of synthetic NPs and enhances the biological functions of OMVs [24]. While OMVs have a membrane barrier, which results in low drug-loading efficiency, NPs have a high drug-loading capacity. When coated with membrane vesicles, nanoparticles exhibit an enhanced drug-loading capacity for cancer therapy [135]. Moreover, membrane vesicle-coated camouflaged NPs can evade the immune system, allowing them to remain in circulation for an extended period. Compared to traditional surface modification of OMVs, this soft modification process is simple and does not require the use of organic solvents, reducing potential limitations and improving biocompatibility [24].

Shi et al. (2020), loaded mesoporous silica nanoparticles (MSNPs) with 5-fluorouracil (5-FU) and camouflaged these drug-loaded nanoparticles with *E. coli*-derived OMVs to improve the ability of MSNPs to target colon cancer. Following oral administration, they observed that this composite delivery system increased cellular uptake by virtue of the mechanism of specific hyaluronic acid receptor-mediated endocytosis and caused a strong cytotoxic effect on cancer cells [139]. This study is a good example that reveals the synergistic effects of a combined OMV-NPs strategy.

A new class of biocompatible hybrid eukaryotic-prokaryotic nanovesicles (EPVs) has been developed, which combines melanoma cytomembrane vesicles for immunotherapy with attenuated *Salmonella* OMVs due to their natural adjuvant properties [140]. The hybrid EPVs were found to trigger a tumor-specific immune response by enhancing DC maturation and T-cell proliferation in melanoma cells. Subsequently, a PLGA-indocyanine green (ICG) core was implanted into EPVs to construct PI@EPV nanoplatforms for photothermal core structure. Under NIR laser irradiation, the PI@EPV surface temperature increased, which facilitated the spread of tumor-specific antigens and adjuvant components as a result of the inducing and photothermal conversion of ICG. This supported the further enhancement of EPV’s anti-tumor immune effects. They found that the PI@EPV and laser combined group exhibited the strongest tumor inhibition activity compared to the control group and other treatment groups. Moreover, the treated mice survived for up to 60 days after the tumor formation.

Zhuang et al. (2022) successfully created new bacteria-plant hybrid nanoplatforms (called BPNs) by integrating thylakoid membranes with OMVs obtained from *E. coli* due to their ability to enhance the antitumor immunity of thylakoid membranes. The effect of OMV-based immunoregulation with photodynamic effects from the thylakoid membrane under laser irradiation provided a strong therapeutic efficacy in colon cancer cells by repolarizing M2 macrophages and increasing the release of tumor-associated antigens presented by dendritic cells, especially effector T lymphocytes that infiltrated the tumor. The combined use of the hybrid structure BPNs and laser irradiation suppressed tumor growth by 94.33% and strongly inhibited tumor metastasis. Furthermore, mice had a 90% survival rate after thirty-six days [123].

Recent research has shown that cancer cell membranes alone are not enough to trigger an antitumor immune response. To address this issue, hybrid membranes have been created by fusing OMVs with cancer cell membranes to facilitate cancer targeting and immunotherapy. Wang et al. (2020) developed a hybrid membrane model (OMV-CC) by fusing cell membranes derived from B16F10 melanoma cells with OMVs derived from *E. coli*, resulting in the specific targeting of melanoma cells in vivo. The researchers further coated hollow polydopamine nanoparticles with this hybrid membrane to form the HPDA@[OMV-CC] structure, which demonstrated selective tumor recognition ability by highly accumulating more in melanoma cells. The structure activated an antitumor immune response by rapidly triggering DC maturation in treated mice, and HPDA-mediated photothermal therapy completely reversed melanoma formation in mice without significant side effects [141]. This study is a useful example of combining the coating- and hybrid-membrane technologies to be used in cancer treatment.

In cancer therapy, the targeted delivery of therapeutics is crucial for effectiveness. To achieve this, microrobots have been engineered for their mobility, functionality, and ability to penetrate tissues and reach cancerous cells [142]. Zhou et al. (2021) developed a combinatorial microrobot system (Motor-OMV) by loading OMVs obtained from *E. coli* onto self-propelled Mg-based micromotors. Intratumoral injection of Motor-OMVs caused significant deterioration of colorectal cancer and melanoma tumor tissues synergistically with tumor tissue destruction caused by the micromotor system and the immuno-stimulating effect of OMVs [143].

### 5.2. Antimicrobial Therapy

#### 5.2.1. Antibacterial Activity

The most common approach to recovering bacterial infections is antibiotic therapy. However, antibiotic treatment cannot be efficient in some cases, e.g., in the case of misuse and self-medication; moreover, it may lead to drug resistance. In this context, the discovery of different antibiotic delivery systems and antibacterial agents is of great importance. The administration of antibiotic therapy is particularly challenging for Gram-negative bacteria due to their double-membrane cell envelope. The ability of BMVs to deliver molecules across the cell envelope of Gram-negative bacteria reveals their antibiotic transport potential [144,145].

The antibacterial effects of OMVs have been documented, with initial evidence provided by Kadurugamuwa and Beveridge in 1996 for lysines in *P. aeruginosa*-derived vesicles [146]. Subsequent studies have identified several other naturally occurring antibacterial molecules in BMVs from different bacteria (Table 4). In *L. acidophilus*, increased amounts of lactacin B in MVs were shown to inhibit the growth of *Lactobacillus delbrueckii* [147]. Moreover, the potential of BMVs as a delivery system for antibacterial agents has been explored as a promising alternative to address the limited availability of such agents at the site of infection [148]. Huang et al. (2020) demonstrated that levofloxacin-loaded OMVs (LEV-OMV) produced by *Acinetobacter baumanii* retained their structure and exhibited a stronger antibacterial effect in mice with an intestinal bacterial infection compared to freely administered levofloxacin [86]. Similarly, BMV-coated nanoparticles (NP-BMV) have been shown to possess antibacterial properties due to the natural immunogenicity of BMVs, as well as providing structural stability and homogeneity. Wu et al. reported high cytokine levels and CRKP-specific antibody production in mice administered with nanoparticles coated with OMVs obtained from carbapenem-resistant *K. pneumoniae* (CRKP) [149]. In a different NP-MV study, Gao et al. showed that antibiotic-loaded PLGA nanoparticles coated with MVs obtained from *S. aureus* were taken up by infected macrophage cells, resulting in antibiotic release and a decrease in the number of *S. aureus* [80].

#### 5.2.2. Antifungal Activity 

Recent studies have shown that, besides their antibacterial properties, BMVs naturally carry antifungal compounds that exhibit antifungal activity. In *Streptomyces albus*, genes encoding the synthesis of antifungal compounds, candicidin and antimycin were identified. The production and secretion of these antifungals were observed in the wild-type and mutant strains of *S. albus*, and it was discovered that *S. albus* MVs carry candicidine [165]. *Lysobacter enzymogenes* produces thermostable antifungal factors (HSAF) and their analogs that are induced by fungal mycelia or chitin-enriched media. The LeLPMO1A gene, which encodes a chitin-binding protein, was identified in the genome analysis of *L. enzymogenes*. Its deletion reduced HSAF production by chitin, indicating the gene’s role in antifungal compound synthesis. In the study on *L. enzymogenes* OMVs, the colocalization of antifungal compounds and the LeLPMO1A gene was established by metabolic and proteomic analyses [166]. Furthermore, OMVs obtained from another strain of *L. enzymogenes* exhibited chitinase and antifungal activity, inhibiting the growth of *Saccharomyces cerevisiae* and *Fusarium subglutinacin* [167]. These findings indicate the potential of BMVs as natural sources of antifungal compounds for the development of antifungal therapeutics.

#### 5.2.3. Antiviral Activity

BMVs can be an intriguing vaccine delivery vehicle for viral antigens because of their self-adjuvant properties and capacity to be adorned with antigens. Finally, studies have evaluated the antiviral abilities of BMVs and their potential as a vaccine platform against viruses. In particular, they have shown promise in protecting against lethal doses of influenza viruses. In a study using BMVs with attenuated lipopolysaccharide content, protection was observed against pandemic H1N1, PR8, and H5N1 virus types, and this antiviral activity was dependent on macrophages, which were induced to produce IFN after BMV administration [168]. Although the immune response generated by BMVs is temporary, a subsequent study applied BMVs with attenuated lipopolysaccharide content and influenza infections consecutively. This resulted in the induction of virus-specific antibodies in mice, and a second infection at week 4 provided complete protection against viral loading in BMV-treated mice for up to 18 weeks [169]. These findings demonstrate the potential of BMVs as a vaccine platform against influenza viruses. 

Other studies have shown that OMVs have potential as a vaccine platform against various viruses, including the Zika virus and severe acute respiratory syndrome coronavirus 2 (SARS-CoV-2). Martins et al. (2018) demonstrated that the administration of OMVs isolated from *Neisseria meningitidis* fused with a Zika virus-infected cell line led to higher antibody production in mice than in non-treated mice [170]. With the increasing course of the COVID-19 epidemic and the formation of variants, alternative vaccine platforms are necessary. The current SARS-CoV-2 vaccines have limitations, and BMVs offer potential advantages. Thapa et al. showed that *Vibrio cholerae* and *E. coli* cells can be genetically modified to secrete OMVs decorated with the receptor binding domain (RBD) of the SARS-CoV-2 spike protein. Mice administered with RBD-decorated OMVs produced antibodies against the spike protein [171]. In another study, BMVs decorated with RBD and loaded with siRNAs against SARS-CoV-2 were shown to suppress infection by specifically targeting lung tissues [172]. Jiang et al. (2022) investigated the efficacy of OMVs against SARS-CoV-2 variants. They administered RBD-decorated OMVs from *S. typhimurium* to mouse models infected with wild-type and delta variants and reported high anti-RBD IgG levels in the blood of all subjects [173].

The use of BMVs as smart drug delivery systems in antimicrobial and anticancer therapies is presented in Figure 5.

## 6. Safety of BMVs as Smart Drug Delivery Systems

The lack of replicative abilities in BMVs makes them less virulent, yet still potentially harmful due to the presence of virulence factors and toxic components [148]. A crucial step for BMV applications is the detoxification of toxic components, particularly LPS found on the surface of BMVs [174]. The LPS consists of acyl chains, core oligosaccharides, lipid A moieties, and O-antigen, with the lipid A moiety being the primary contributor to LPS toxicity [144]. Detergents are often used to remove LPS from BMVs, which reduces their toxicity. Another approach involves editing genes in bacterial cells to reduce toxicity, such as inactivating the msbB gene or modifying operon genes involved in core oligosaccharide and O-antigen synthesis [93,125]. However, genetic modification can also result in decreased BMV yield and needs optimization. An alternative approach is modifying the phosphorylation of the lipid A portion, such as expressing the Hp0021 gene in *E. coli* to obtain monophosphorylated lipid A. While promising, this approach has yet to be used in BMV biotechnological applications [144,175]. Additionally, selecting non-pathogenic bacteria to obtain BMVs with low toxicity is an alternative approach. For instance, BMVs from symbiotic bacteria can stimulate immune regulation and prevent colitis [176].

Apart from LPS, many virulence factors, such as cytotoxins and adhesins carried by BMVs, also cause toxicity. In a study conducted to eliminate this toxicity, a mutant strain was created by deleting 14 genes encoding virulence factors in *P. aeruginosa*. BMVs produced from the mutant strain did not cause any death when administered to mice. Time and applicability are limiting factors in this detoxification approach [177].

## 7. Obstacles of BMVs for Clinical Use as Smart Drug Delivery Systems

We have reviewed the properties of BMVs and their various applications. While BMVs have demonstrated their effectiveness against cancer and infectious diseases in both in vitro and in vivo studies, there are still unresolved practical issues in their clinical use as potential smart delivery vehicles. It is crucial to address these issues to bridge the gap between laboratory research and clinical applications of BMVs [60].

Biosafety. The use of BMVs in clinical practice faces a major obstacle, which is safety. OMVs are typically derived from Gram-negative bacteria and contain endotoxins and virulence factors such as LPS that can elicit immune responses and toxicity. To address this challenge, genetic engineering has been used to develop OMVs with reduced/attenuated endotoxins by deleting the msbA, msbB, lpxM, and lpxL1 genes [55]. LPS-deficient OMVs exhibit lower immunogenicity than those with normal LPS levels [178], but the optimal balance between low toxicity and high immunogenicity remains a challenge. Additionally, to ensure safety in clinical trials, OMVs’ endotoxin levels should be as low as possible, and further in vivo studies are necessary to expedite OMV applications in humans.

Strain selection. OMVs obtained from *E. coli* are widely used in cancer therapy (Table 2). However, in using BMVs for these purposes, the identification and selection of specific strains can provide a better balance between the safety and immunogenicity of BMVs. Thus, there is a need for methods to control the immunotoxicity of BMVs to increase their biosafety. 

Biogenesis. Literature reports suggest that information on MVs is relatively limited compared to OMVs [48]. Consequently, the mechanisms of BMV biogenesis remain poorly understood. Further research into the underlying biogenesis mechanisms will aid in discovering and characterizing membrane vesicles from other bacterial species. This will also facilitate the development of improved bacterial-based nanoplatforms for biomedical applications through more in vivo and in vitro studies.

Standardized Production. Obtaining BMVs involves many methods, and there is no standardized analytical protocol. Different production and isolation techniques result in a large heterogeneity of the obtained BMV populations due to the characteristics of BMVs, such as their size and components. This makes it difficult to achieve reproducibility and consistency of results in clinical applications. The interaction of heterogeneous BMVs with target cells and their intracellular fate is unpredictable. Therefore, the development of methods like proteomics and lipidomics to produce homogeneous BMV structures and analyze their composition is critical for their use in clinical applications [135]. These methods will pave the way for the successful use of BMVs for clinical purposes. 

Scalability. The collection and purification of BMVs using the ultracentrifugation method require advanced equipment and qualified personnel, making it time-consuming and costly [131]. Furthermore, commercial kits available for BMV purification suffer from low yield and purity issues, thereby causing high prices [179]. Thus, there is an urgent need to develop new and scalable BMV separation/purification methods that can continuously separate BMVs from the culture medium with high yield and purity. Although bacteria can be easily produced in large quantities using large fermentation vessels, the amount of BMVs released from bacteria is not sufficient for their mass production to be cost-effective. Therefore, there is a requirement for high-efficiency production of BMVs to be scalable for clinical applications. To achieve this, different cultural conditions and systems must be explored to optimize BMV production and lay the foundation for BMV industrialization. 

Drug-loading. One of the challenges of loading therapeutics into BMVs is the bilayer lipid membrane, which makes it difficult despite various drug-loading strategies. This low loading rate has hindered the clinical application of BMVs. Although Gao et al. performed drug loading with a pH gradient to overcome this situation, they were only able to increase drug loading to 12% [180]. As a result, new approaches and methods should be developed to increase drug loading to BMVs.

The limitations of using BMVs in clinical practice as previously mentioned are concerning. To address this limitation, an alternative method is the development of controllable artificial BMVs that enable a more comprehensive understanding of their active ingredients. This can be achieved by obtaining detoxified BMVs, which can be exploited for their unique biological properties in biomedical applications [130].

## 8. Outlook and Conclusions

With the publication of research over the last two decades, our knowledge of the biogenesis and function of BMVs has advanced rapidly. BMVs are among the most extensively studied bacterial strategies in the fight against infectious and non-infectious diseases. In this review, we discussed the many unique properties of BMVs, especially OMVs generated from Gram-negative bacteria, their current production and purification procedures, characterization methods, and their functions in biomedical applications. In addition to their direct production by bacterial cell culture, biotechnological developments have enabled OMVs to be customized to contain various proteins and ligands using recombinant DNA technology to produce them in accordance with the desired purpose, and thus their usability as carrier systems for antigens and therapeutic agents. We have clarified in detail the strengths and weaknesses of smart drug delivery systems in this review, which serve as the foundation for the use of BMVs in biomedical applications. In addition, OMVs can be functionalized not only as targeted drug delivery systems but also as bioimaging platforms using protein–protein binding pairs and multi-domain scaffold structures through genetic modification. 

To date, BMVs have been researched and encouraging advances have been made for the transmission of many therapeutic cargoes (chemotherapeutic drugs, therapeutic nucleic acids, antibiotics, nanoparticles, proteins, etc.). However, despite the many notable studies that have been successful in vitro and in vivo with sophisticated approaches to smart drug and/or cargo delivery, the critical limitations detailed here need to be overcome in order to pave the way for their clinical use, and more studies are needed to do so. Limited research is available on the biogenesis of BMVs and, in particular, MVs from Gram-positive bacteria, and future studies in this field will aid discovery and characterization of membrane vesicles from bacterial species. The fact that researchers will utilize multiomic techniques to disclose the composition variety and diverse nature of BMVs using distinct datasets will provide us greater insight into their diversity of functions. In particular, the safety of OMVs is a major issue and the identification of different strains from *E. coli* and new ways of weakening endotoxin content are critical to achieving a balance in terms of immunogenicity and the safe use of OMVs. In this context, more clinical and in vivo research are needed to investigate bacterial strains and evaluate immunogenicity. Moreover, the optimization and development of standard and repeatable protocols in the production and purification of BMVs are crucial both for their production and for their functionalization on a large scale. The main problem as carrier systems is the low loading rate of therapeutical agents, since BMVs contain bilayer lipid barriers. In order to overcome this, researchers turn to the BMV’s production strategy designed with stimuli, but it is thought that studies on the production and design of controllable artificial BMVs will be more solution-oriented in order to increase the loading capacity of the therapeutic agent.

Overall, the door to a new era has been opened in the use of bacteria-based nanotechnology for the treatment of diseases. The use and development of BMVs as nanocarrier smart systems in the treatment and diagnosis of target cells is the focus of studies that will contribute to the development of new bacterial-based nanovesicle technology. 

## Figures and Tables

**Figure 1 pharmaceutics-15-01052-f001:**
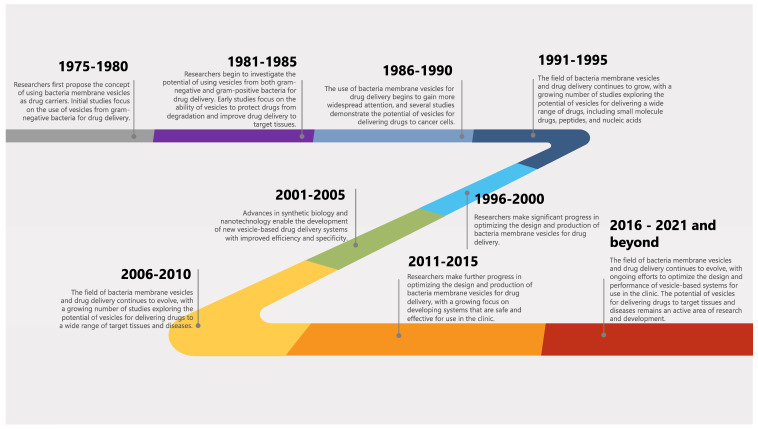
A general timeline of key events and milestones in the history of BMVs and their applications in drug delivery.

**Figure 2 pharmaceutics-15-01052-f002:**
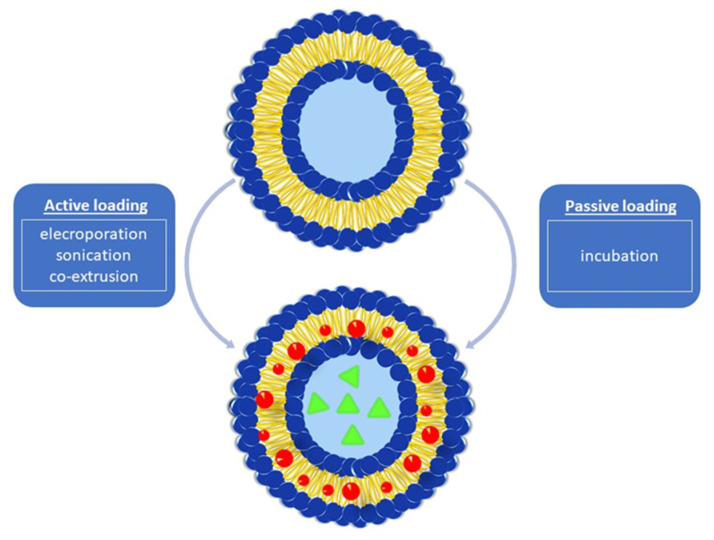
Drug loading approaches into BMVs.

**Figure 3 pharmaceutics-15-01052-f003:**
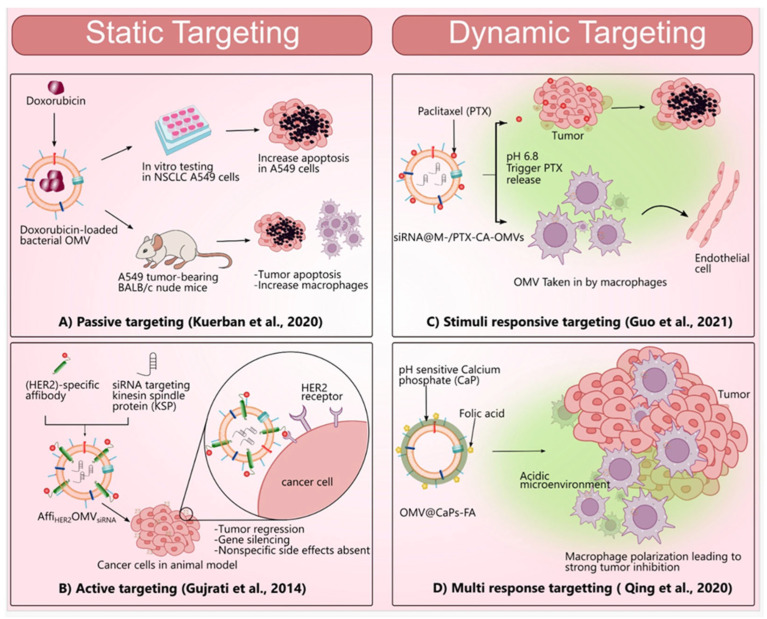
Static and dynamic targeting strategies of BMVs used as carrier systems [77,82,87,88].

**Figure 4 pharmaceutics-15-01052-f004:**
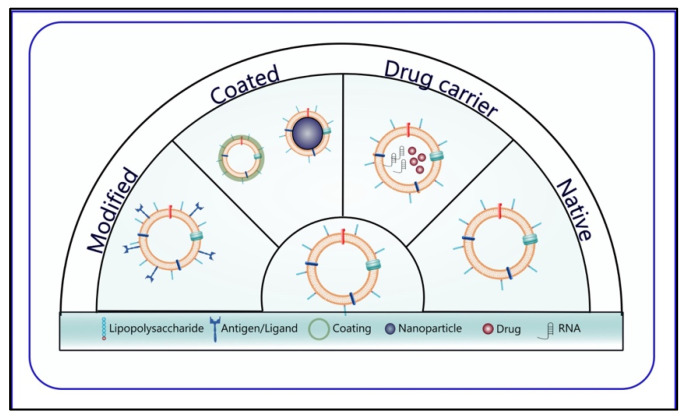
Different OMV approaches used in cancer therapy.

**Figure 5 pharmaceutics-15-01052-f005:**
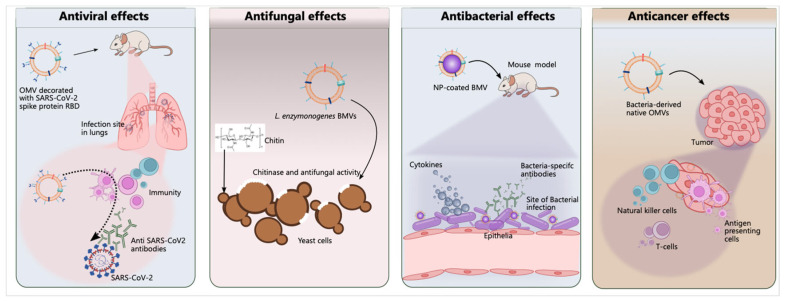
Potential uses of BMVs in antimicrobial and anticancer therapy.

**Table 1 pharmaceutics-15-01052-t001:** The classification of SDDSs as well as their advantages and disadvantages.

Classification of SDDS		Advantages	Disadvantages
Organic	Polymeric nanoparticles	Controllable particle and surface properties Enhanced stability of API Tunable release properties API encapsulation both hydrophilic and lipophilic character	Difficulty in adapting production processes on an industrial scale Residual material associated with production process
	Hydrogels	Ability to absorb water or biological fluids Capacity to mimic biological structures with 3D structure Biodegradable and nontoxic nature Site-specific application	Possibility of drug deactivation during production Limited hydrophobic drug delivery Difficulty in sterilization Mechanically unstable High production costs
	Dendrimers	Increased drug solubility High loading efficiency through internal cavities Surface modification with terminal functional groups Enhanced permeation effects and long in vivo circulation lifetime	High nonspecific toxicity Increased cytotoxicity due to an increase in the number of generations Possible hemolytic activity Relatively expensive raw material requirement
	Liposomes	Nontoxic, biodegradable, and flexible vesicles cell membrane-like structure Simultaneous entrapment of hydrophilic and lipophilic API prolonging circulation time Possible to formulate sterically stabilized liposomes	Possibility of organic solvent residue Lipid oxidation/hydrolysis problem during shelf life
	Lipid nanoparticles	Safe composition of physiological lipids Avoiding the use of organic solvents Economical and low-cost production Possible to scale up Site-specific drug delivery	Low drug-loading capacity for hydrophilic molecules Expulsion of API due to polymorphic transitions during storage Burst release
Inorganic	Carbon nanotubes	Easy to modify and functionalize Sequential structure High mechanical strength Effective for molecules to enter cells	Highly hydrophobic nature The lack of solubility in solvents compatible with biological fluid Biodegradation problem
	Mesoporous silica system	High surface–volume ratio Presence of nanopores Low-cost complex system design Avoiding the early drug release	Solubility and biodegradability characteristics Presence of silanol groups and their interaction with membrane lipids
	Metal nanoparticles	Availability of green synthesis Tunable size, geometry, and surface Suitable for large-scale production Unique optical, electronic physicochemical features Can be used as a diagnostic tool	Not biodegradable Tendency to accumulate non-specifically in the body Environmental toxicity risk
Biologic	Exosomes	Ability to mediate intercellular communication Resistant to digestive enzymes	Challenges related to the isolation and purity Rapid elimination from the bloodstream Limited large-scale production
	Bacterial membrane vesicles	Easy-to-access raw material source Able to be designed with the help of genetic engineering Surface modification with biological ligands	Limited scalable manufacturing Relatively low BMV yield Lack of standardization in production

**Table 2 pharmaceutics-15-01052-t002:** Ligands used for active targeting of BMVs in cancer research.

Class	Indication	Ligand	Target	References
Aptamer	Breast cancer	AS-1411	Nucleolin	[103]
Carbohydrate	M2 macrophages	Mannose	Mannose receptor (MR, CD206+)	[87]
Peptide	Melanoma	RGP and RGD	αvβ3 integrin	[96]
		RGP		[104]
	Breast cancer Melanoma Colon cancer	LyP1	p32/gC1qR	[105,106]
Protein	Breast cancer Ovarian cancer	HER2 affibody	HER2 receptor	[77]
	Breast cancer	EGFR affibody	EGFR	[107]
	Colon cancer	PD1	PD-L1	[108]
Small molecule	Breast cancer	Folic acid (FA)	Folate receptor (FR)	[88]

**Table 3 pharmaceutics-15-01052-t003:** Application of OMVs as smart delivery systems in cancer therapy.

Bacterial Source of OMVs	Modification and/or Guest Molecules	Particle Size (nm)	Modification or Loading Method	Stimulating Factor	Therapy Strategy	Outcomes	References
*E. coli* DH5α	Ce6	70–140	Co-incubation	Photodynamic	- trigger drug release with laser irrigation - achieve effective antitumoral effects with combined photo/chemo/immunotherapy	- decreased cell viability - induced ROS generation - suppression of tumor growth and eradication of tumor tissues - shifting macrophages M2-to-M1 polarization in 4T1 breast cancer-bearing mice in vivo	[133]
	DOX						
*E. coli* BL21 (ΔmsbB)	tRNALys-pre-miRNA-126	108.2	Genetic engineering	-	Targeting breast cancer cells by specific binding of the aptamer to nucleolus proteins on the surface of breast cancer cell membranes.	- inhibition of cell proliferation - decreased expression of target genes corresponding to invasion miRNA - accumulation in tumor tissues - inhibition of tumor growth	[103]
	aptamer AS1411		Incubation				
Attenuated *Salmonella*	αPD-L1	140.907	Extrusion	Photodynamic	To increase the amount of O_2_ in tumor cells by means of negatively charged catalase and Ce-6, thus overcoming the hypoxia barrier in front of the photodynamic effect and obtaining an effective antitumoral effect.	- accumulation in tumor tissues - decreased cell viability - decreased HIF-1α expression - induced ROS generation and increased dissolved O_2_ - maturation of dendritic cells - increase in effector T cells population in 4T1 breast cancer-bearing mice in vivo	[114]
	Catalase-Ce6		Co-extrusion				
*E. coli* K12 (ΔmsbB)	LyP1 polypeptide	~136.9	Genetic engineering	-	- effective tumor targeting and increased internalization rate through the LyP1 polypeptide - ensuring the expression of PD1 in tumor cell membranes and self-blocking of the PD1/PD-L1 axis in tumor cells	- inhibition of tumor growth inhibition - decrease in Ki-67+ cell populations - increase of apoptotic cells in tumor tissues - increase of NK cells and CTLs population - maturation of dendritic cells - increase of TCM cells	[105]
	PD1 plasmid		Electroporation				
*E. coli* Nissle 1917	CuS	170.2 ± 0.2	Incubation	Photothermal	Generating strong hyperthermia in tumors through the photothermal effect.	- effective photothermal conversion - induce stronger cytotoxicity - effectively accumulate in tumor tissues - induce ICD of tumor cells - remarkable tumor growth inhibition - maturation of dendritic cells - decrease in Ki-67+ cell populations - activation of cytotoxic T cells in 4T1 breast cancer-bearing mice in vivo	[113]
*E. coli* BL21 (ΔmsbB)	Redd1 siRNA	130 ± 15.16	Electroporation	pH-sensitive	- siRNA-mediated silencing of key genes in cancer cells - ensuring the pre-release of the therapeutic agent paclitaxel by triggering pH-sensitive CA	- downregulated the mRNA expression level of Redd1 - inhibition of malignant cell proliferation - effective metastasis inhibition capacity - increased cleaved caspase-3 - decreased Ki-67+ cell populations - increased effector T cells and decreased Treg cell population in 4T1 breast cancer-bearing mice in vivo - maturation of dendritic cells	[87]
	DSPE-PEG-CA-PTX		Co-incubation				
*E. coli* (ΔmsbB/ΔpagP)	GALA	135.76 ± 30.33	Genetic engineering		Ensuring targeted binding, specifically to cancer cells overexpressing the EGF receptor through expressed affi-EGFR proteins.	- specific binding to EGFR-overexpressing breast cancer cells - no cytotoxicity - no immunogenicity	[107]
	EGFR						
*E. coli* DH5α	BFGF	166.9	Genetic engineering	-	To achieve a lasting and effective antitumor effect by inducing the production of anti-BFGF autoantibodies.	- best tumor suppression function because of induction of anti-BFGF autoantibodies - decreased tumor volume and inhibition of tumor growth - inhibition of FGFR1 phosphorylation level - reduction of lung metastasis - increased caspase-3 expression - decreased Treg cells and increased Th1 and CTLs in the mouse B16F10 xenograft model	[128]
*E. coli*	TRAIL	94.46 ± 5.22	Genetic engineering	Photothermal	- specific tumor targeting by the ligand RGP or RGD peptides - increasing the synergistic anti-tumor therapeutic effect by combining photothermal therapy	- effective transdermal efficiency - effective tumor targeting by peptide ligand - accumulation in melanoma spheroids - induction of apoptosis in tumor cells	[96]
	ICG		Incubation				
	RGP or RGD peptide		Incubation				
*E. coli* (ΔmsbB)	PD1	32.7 ± 10.6	Genetic engineering	-	Enabling both internalizations of OMVs by binding of PD1 to PD-L1 on the surface of tumor cells as well as preventing inhibition of T cell proliferation by tumor cells through inhibition of PD-L1.	- maturation of dendritic cells - accumulate in tumor tissues - induction of apoptosis and necrosis in tumor cells - increase of effector memory T cells, NK cells and macrophages population in CT-26 colon cancer-bearing mice in vivo	[108]
Attenuated *K. pneumonia* ATCC 60095	DOX	93.09	Incubation	-	Enabling the generation of chemoimmunotherapeutic responses by using OMVs as drug delivery systems for chemotherapeutic agents.	- cytotoxic effect on lung cancer cells - increased cleaved caspase-3 and PARP - triggering extensive tumor necrosis - increase in tumor growth inhibition - increased serum TNF-α and IL-6 levels	[82]
*E. coli*	TRAIL	94.54 ± 1.46	Genetic engineering	Photothermal	- specific tumor targeting by the ligand RGP peptide - increasing the synergistic anti-tumor therapeutic effect by combining photothermal therapy	- infiltration and accumulation of OMVs in tumor spheroid - inhibition of tumor cell proliferation and invasion - induction of apoptosis in tumor cells - decrease in c-FLIP and survivin protein expression - increase in cleaved caspase-3 and caspase-8 protein expression - effective skin penetration - increase in temperature of tumor sites	[104]
	ICG		Electrostatic interaction				
	RGP peptide		Incubation				
*E. coli* BL21	Calcium phosphate (CaP) shells	100–150	Incubation	pH-sensitive	- providing tumor inhibition by dissolving CaP shells in the tumor environment with a slightly acidic pH - increasing the synergistic anti-tumor therapeutic effect by combining photothermal with immunotherapy	- accumulate in tumor tissues - pronounced tumor inhibition - increased effector T lymphocyte and decreased Treg cell population in CT-26 colon cancer-bearing mice in vivo - increased cytokine secretion levels in CT-26 colon cancer-bearing mice in vivo - induced ICD of tumor cells - maturation of dendritic cells	[88]
	ICG			Photothermal			
*E. coli* K12 (ΔmsbB)	Melanin	20–100	Genetic engineering	Photothermal	Both to create contrast for optoacoustic imaging on cancer cells and to achieve an anti-tumoral effect benefiting from the high photothermal conversion effect of melanin.	- effective photothermal conversion - accumulation of tumor tissues via the EPR effect - increased tumor surface temperature - decreased tumor growth - necrosis in tumor tissues	[93]
*E. coli* DH5α	Protein E7 (HPV16E7)	20–200	Genetic engineering	-	Enabling antitumoral effects to occur by stimulating cellular immune responses of antigen-presenting recombinant OMVs.	- maturation of dendritic cells - significant suppression of tumor growth	[97]
*E. coli* K12 (ΔmsbB)	HER2	30–250	Genetic engineering	-	- specific tumor targeting by the HER2 affibody - siRNA-mediated silencing of key genes in cancer cells	- inhibition of cell proliferation through siRNA-mediated degradation of KSP mRNA - large accumulation in tumor tissues	[77]
	KSP siRNA		Electroporation				

**Table 4 pharmaceutics-15-01052-t004:** Antibacterial effect studies of BMVs.

BMV Source	Application Type of BMVs	Active Ingredient	Target Bacteria	References
*P.aeruginosa*	Natural Drug delivery	Autolysin Gentamicin	*S. aureus* *E. coli*	[150,151]
*Lysobacter* sp. XL1	Natural	Endopeptidase L5	*S. aureus* *Erwinia marcescens*	[152]
*Myxococcus xanthus*	Natural	Hydrolase content	*E. coli*	[153]
*Cystobacter velatus* Cbv34, *Cystobacter ferrugineus* Cbfe23	Natural	Cystobactamid	*S. aureus* *E. coli*	[154,155]
*Lysobacter capsici*	Natural	Bacteriolytic enzymes	*Micrococcus roseus* *S. aureus* *Micrococcus luteus* *Bacillus cereus*	[156]
*Burkholderia thailandensis* E264	Natural	Peptidoglycan hydrolases, 4-hydroxy-3-methyl-2-(2-non-enyl)-quinoline (HMNQ), long-chain rhamnolipid	*A. baumannii* *S. aureus*	[157]
*L. acidophilus*	Natural	lactacin B	*L. delbrueckii*	[147]
*Lacticaseibacillus casei* BL23	Natural	Antibofilm agent peptidoglycan hydrolases	*Salmonella enterica*	[158]
*Buttiauxella agrestis*	Drug delivery	Gentamicin	*Buttiauxella agrestis*	[159]
*A. baumannii*	Drug delivery	Levofloxacin, Amikacin Ciprofloxacin, Norfloxacin	*K. pneumoniae* *E. coli* *P. aeruginosa*	[86]
*Shigella flexneri*	NP-OMV	Poly(anhydride) NP-OMV	*Shigella flexneri*	[160]
*E. coli*	NP-OMV	Gold nanoparticles (AuNPs)-OMV	Unknown	[161]
*Vibrio cholerae*	NP-OMV	Chitosan-tripolyphosphate NP-OMV	*Vibrio cholerae*	[162]
*Helicobacter pylori*	NP-OMV	PLGA NP-OMV	*Helicobacter pylori*	[163]
*S. aureus*	NP-MV	PLGA NP-MV	*S. aureus*	[80]
*K. pneumoniae*	NP-OMV	-	carbapenem-resistant *K. pneumoniae*	[149]
*Bordetella bronchiseptica*	NP-OMV	Glycyrrhizic acid-NP	*Bordetella bronchiseptica*	[164]

## Data Availability

Not applicable.

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
