# Peer review of "Bacterial Membrane Vesicles as Smart Drug Delivery and Carrier Systems: A New Nanosystems Tool for Current Anticancer and Antimicrobial Therapy"

_pharmaceutics, 2023, doi:10.3390/pharmaceutics15041052_

Round 1

Reviewer 1 Report

Referee Report

Title: Bacterial Membrane Vesicles as Carrier Adjuvant Systems in Smart Drug Delivery

Manuscript ID: pharmaceutics-2262327

By Celik et al

Submitted to Pharmaceutics (ISSN 1999-4923)

Comments

This is a review about the application of BMV as a smart drug carrier in theranostic. This manuscript is quite thorough and I only have some minor comments:

1.       Title: From the title, it is understood that BMV is used in the drug delivery but the authors did not say for what? For example, for medical imaging, cancer therapy, or theranostic?

2.       Abstract: From the abstract, I cannot see the aim of this review. Generally, a central question is found, resulting in conducting this review. It is not clear why the authors wanted to submit this work.

3.       Introduction: The authors used a lot of paragraphs to explore the background and history of drug delivery system. Only the last paragraph is related to BMV.

4.       Introduction, L73-83: When mentioning drug delivery using nanomaterials, please use more updated references such as Siddique et al (Nanomaterials 2022;12:2826) and Siddique et al (Nanomaterials 2020;10:1700.)

5.       Section 2: It is good, if possible, to provide a schematic diagram showing the procedure and mechanism showing how BMV can carry the drug to the cell.

6.       Section 3: The “smart” approach of BMV (i.e. dependence of pH and temperature) can also be found in other functionalized nanomaterials in drug delivery. So, can the authors mention more how the BMV is better or different from other smart drug carrier options?

7.       Figure 1 looks like a picture rather than a timeline chart.

8.       There are two Section 3.4.2 (L427 and L473).

9.       Section 4: This section is too short. In fact, this section should be linked to medical imaging using BMV as a contrast agent. The authors may want to expand this section.

1.   Section 8: The authors may want to mention more the future prospectives in this field and what should be done more after the review, for example, to conduct clinical trials.

Author Response

DETAILED RESPONSES TO THE REVIEWER' COMMENTS

Ref. no:  Pharmaceutics-2262327

Title: Bacterial Membrane Vesicles as Carrier Adjuvant Systems in Smart Drug Delivery

Authors: Pinar Aytar Celik, Kubra Erdogan Gover, Dilan Barut, Blaise Manga Enuh, Gulin Amasya, Ceyda Tuba Sengel-Turk, Burak Derkus, Ahmet Cabuk.

Article Type: Review paper

Dear Editor,

Thank you and the reviewer for their useful comments and suggestions on the structure of our manuscript. We have applied the corrections suggested by the reviewer and the details are listed below point by point. Where corrections have been made in the manuscript's main text, the text has been colored red, and references to the line have been provided in this document.

Reviewer #1: This is a review about the application of BMV as a smart drug carrier in theranostic. This manuscript is quite thorough and I only have some minor comments:

Comment 1. Title: From the title, it is understood that BMV is used in the drug delivery but the authors did not say for what? For example, for medical imaging, cancer therapy, or theranostic?

Response 1. The authors thank the reviewer for this valuable suggestion. The title of the review has been edited in line with the reviewer’s recommendation as follows.

“Bacterial Membrane Vesicles as Smart Drug Delivery and Carrier Systems: A New Nanosystems Tools for Current Anticancer and Antimicrobial Therapy”

Comment 2. Abstract: From the abstract, I cannot see the aim of this review. Generally, a central question is found, resulting in conducting this review. It is not clear why the authors wanted to submit this work.

Response 2. We thank the reviewer for the valuable comment. In line with the reviewer's recommendation, the abstract section was edited, and the purpose of the relevant review was stated as follows, especially in the last part.

“In conclusion, this review paper aims to provide a comprehensive overview of the state-of-the-art field of BMVs as SDDS, encompassing their design, composition, fabrication, purification, and characterization, as well as the various strategies used for targeted delivery. Considering this information, it is aimed to provide researchers in the field with a comprehensive understanding of the current state of BMVs as SDDS, enabling them to identify critical gaps and formulate new hypotheses to accelerate the progress of the field.”

Comment 3. Introduction: The authors used a lot of paragraphs to explore the background and history of drug delivery system. Only the last paragraph is related to BMV.

Response 3.    We thank the reviewer for the comment. Based on reviewer 1’s comment, the introduction has been updated and more updated references were added.

Comment 4. Introduction, L73-83: When mentioning drug delivery using nanomaterials, please use more updated references such as Siddique et al (Nanomaterials 2022;12:2826) and Siddique et al (Nanomaterials 2020;10:1700.)

Response 4.  We thank the reviewer for the comment. Based on reviewer 1’s comment, the introduction has been updated and more updated references were added.

Comment 5. Section 2: It is good, if possible, to provide a schematic diagram showing the procedure and mechanism showing how BMV can carry the drug to the cell.

Response 5.  We thank the reviewer for this valuable suggestion. The authors did not address the host cell entry mechanisms of BMVs, since the subject of the review is quite comprehensive and long. Therefore, no visual has been drawn regarding this issue. In order not to disrupt the subject flow of the current review, the mechanisms of BMV entry into the host cell are the subject of a separate review and can be explained in more detail in another review.

Comment 6. Section 3: The “smart” approach of BMV (i.e. dependence of pH and temperature) can also be found in other functionalized nanomaterials in drug delivery. So, can the authors mention more how the BMV is better or different from other smart drug carrier options?

Response 6. The authors thank the reviewer for the comment. The authors have explained the advantages of BMVs over other carrier systems by adding a detailed Table 1.

Comment 7. Figure 1 looks like a picture rather than a timeline chart.

Response 7. The authors thank the reviewer for the comment. The timeline is visualized again.

Comment 8. There are two Section 3.4.2 (L427 and L473).

Response 8. We thank the reviewer for pointing out this issue. The subtitle of Section 3 has been corrected in L434 as follows.

“3.4.1. Active cargo loading”

Comment 9. Section 4: This section is too short. In fact, this section should be linked to medical imaging using BMV as a contrast agent. The authors may want to expand this section.

Response 9. We thank the reviewer for the valuable comment. Currently, there is limited research in the literature on the use of BMVs as biosensing and bioimaging tools. In line with the reviewer's comment, a new study numbered as 120 has been added to this section (L755-768), and a study numbered as 91 has been re-explained in detail. BMVs loaded with melanin as a contrast enhancer for medical imaging were mentioned in study 91 (L725-741).

Comment 10. Section 8: The authors may want to mention more the future prospectives in this field and what should be done more after the review, for example, to conduct clinical trials.

Response 10. The authors thank the reviewer for this valuable suggestion. This section has been edited and rewritten in line with the reviewer's comment.

Reviewer 2 Report

The article “Bacterial Membrane Vesicles as Carrier Adjuvant Systems in Smart Drug Delivery”. This review paper aims to provide a comprehensive overview of the state-of-the-art in the field of BMVs as smart drug delivery systems, encompassing their design, composition, fabrication, purification, and characterization, as well as the various strategies used for targeted delivery. This article has clear thinking and reasonable structure. Author have clarified in detail the strengths and weaknesses of smart drug delivery systems. 

Here are my suggestions for this article.

The design and classification of intelligent drug delivery systems can be classified in a table to make the content more scannable.

Author Response

DETAILED RESPONSES TO THE REVIEWER' COMMENTS

Ref. no:  Pharmaceutics-2262327

Title: Bacterial Membrane Vesicles as Carrier Adjuvant Systems in Smart Drug Delivery

Authors: Pinar Aytar Celik, Kubra Erdogan Gover, Dilan Barut, Blaise Manga Enuh, Gulin Amasya, Ceyda Tuba Sengel-Turk, Burak Derkus, Ahmet Cabuk.

Article Type: Review paper

Dear Editor,

Thank you and the reviewer for their useful comments and suggestions on the structure of our manuscript. We have applied the corrections suggested by the reviewer and the details are listed below point by point. Where corrections have been made in the manuscript's main text, the text has been colored red, and references to the line have been provided in this document.

Reviewer #2: The article “Bacterial Membrane Vesicles as Carrier Adjuvant Systems in Smart Drug Delivery”. This review paper aims to provide a comprehensive overview of the state-of-the-art in the field of BMVs as smart drug delivery systems, encompassing their design, composition, fabrication, purification, and characterization, as well as the various strategies used for targeted delivery. This article has clear thinking and reasonable structure. Author have clarified in detail the strengths and weaknesses of smart drug delivery systems. 

Here are my suggestions for this article.

Comment 1. The design and classification of intelligent drug delivery systems can be classified in a table to make the content more scannable.

Response 1.  The authors thank the reviewer for this valuable suggestion. A detailed table (Table 1) has been added for the classification of SDDS in line with the reviewer’s suggestion.

Reviewer 3 Report

Here, authors systemically described the potential use of bacterial membrane vesicles as the drug carrier for the smart drug delivery. The manuscript was written in a logical manner. Thus, I suggest the acceptance after minor revision.

Some issues should be addressed.

1.     In section 4 and 5, some important studies should be described in details and demonstrated in illustration if necessary.

2.     In the “Introduction” section, paragraph 2 and 3 were too lengthy. The description of “drug delivery system” seems a well-known and acceptable concept for the readers in this area.

3.     In section 5.1, the application of BMVs demonstrated great advantages, particularly their tumor-homing ability as well as the induction of the local immunity for cancer therapy. These two features should be introduced in details.

4.     Moreover, since cell- derived vesicles including exosomes also exhibited the similar properties as BMVs, what is the advantage of BMVs? I suggest the comparison between them should be claimed, which is necessary to highlight the potential of BMVs.

5.     In the section 8, it is recommended to fully summarize the work of the manuscript and also encouraged to present the authors’ own unique insights on the future development of BMVs.

6.     Is there any study on the immunity of BMVs, since the major concern of the BMVs is whether they triggered the immunity for in vivo application? Authors should clarify this in details.

Author Response

DETAILED RESPONSES TO THE REVIEWER' COMMENTS

Ref. no:  Pharmaceutics-2262327

Title: Bacterial Membrane Vesicles as Carrier Adjuvant Systems in Smart Drug Delivery

Authors: Pinar Aytar Celik, Kubra Erdogan Gover, Dilan Barut, Blaise Manga Enuh, Gulin Amasya, Ceyda Tuba Sengel-Turk, Burak Derkus, Ahmet Cabuk.

Article Type: Review paper

Dear Editor,

Thank you and the reviewer for their useful comments and suggestions on the structure of our manuscript. We have applied the corrections suggested by the reviewer and the details are listed below point by point. Where corrections have been made in the manuscript's main text, the text has been colored red, and references to the line have been provided in this document.

Reviewer #3: Here, authors systemically described the potential use of bacterial membrane vesicles as the drug carrier for the smart drug delivery. The manuscript was written in a logical manner. Thus, I suggest the acceptance after minor revision.

Some issues should be addressed.

Comment 1. In section 4 and 5, some important studies should be described in details and demonstrated in illustration if necessary.

Response 1. We thank the reviewer for the valuable comment. In line with the reviewer's comment, a new study numbered as 120 has been added to this section (L755-768). Since Section 5 is supported in detail by Table 2 as well as the text, no changes have been made in this section.

Comment 2. In the “Introduction” section, paragraph 2 and 3 were too lengthy. The description of “drug delivery system” seems a well-known and acceptable concept for the readers in this area.

Response 2. The authors thank the reviewer for this valuable suggestion. Based on reviewer 2’s comment, the introduction has been updated.

Comment 3. In Section 5.1, the application of BMVs demonstrated great advantages, particularly their tumor-homing ability as well as the induction of the local immunity for cancer therapy. These two features should be introduced in details.

Response 3. The authors thank the reviewer for this valuable suggestion. The local immune-enhancing action way of BMVs in cancer therapy is explained in L772-777. In addition, the findings are also presented in Table 2.

It is extensively explained that the tumor-homing capabilities of BMVs can be found in subsection 5.1.3 on the modification of BMVs with tumor antigens and ligands to achieve specific tumor targeting, and in subsection 5.1.4 on the coating and fusion of BMVs with nanoparticles, cancer cell membranes.

Comment 4. Moreover, since cell-derived vesicles including exosomes also exhibited the similar properties as BMVs, what is the advantage of BMVs? I suggest the comparison between them should be claimed, which is necessary to highlight the potential of BMVs.

Response 4. We thank the reviewer for the valuable comment. The advantages of BMVs compared to exosomes are described in the last part of subsection “2.2. Classification” as follows.

“Exosomes from eukaryotic cells and membrane vesicles from bacteria have numerous structural and biological functional similarities [24]. Although exosomes are being investigated more and more as carrier systems with high biocompatibility and translation ability around the world, their manufacture is challenging and costly, especially in large-scale mammalian cultures, when compared to BMVs. On the other hand, the production of BMVs in bacterial cultures is faster, simpler, and relatively cheaper. In addition, BMVs are nanostructures specifically suitable for design and production, where genetic engineering is more convenient [11]. Although research on exosomes in biological applications is intense, investigations in this field have increased steadily in recent years after the discovery of BMVs [25].”

Comment 5. In the section 8, it is recommended to fully summarize the work of the manuscript and also encouraged to present the authors’ own unique insights on the future development of BMVs.

Response 5. The authors thank the reviewer for this valuable suggestion. This section has been edited and rewritten in line with the reviewer's comment.

Comment 6. Is there any study on the immunity of BMVs, since the major concern of the BMVs is whether they triggered the immunity for in vivo application? Authors should clarify this in details.

Response 6. We thank the reviewer for the valuable comment. The findings that BMVs induce immunity are given in “the outcomes column” in Table 2, and this information is not given again in the text to avoid duplication. However, this column has been rearranged to clearly understand the information that immunity has been investigated in animal experiments, and it has been emphasized that these results were made in vivo studies. If the reviewer refers to the biosafety of BMVs as a major concern, an in vivo study is mentioned in L725-737 regarding this issue.

Round 2

Reviewer 1 Report

I am satisfied with the modifications and corrections made by the authors as per my comments. I also accepted the responses from my concerns in some sections.